# ReactDance: Hierarchical Representation for High-Fidelity and Coherent Long-Form Reactive Dance Generation

**Jingzhong Lin**[1], **Xinru Li**[1], **Yuanyuan Qi**[1], **Bohao Zhang**[1], **Wenxiang Liu**[1], **Kecheng Tang**[1], **Wenxuan Huang**[1], **Xiangfeng Xu**[1], **Bangyan Li**[1], **Changbo Wang**[1], **Gaoqi He**[1]✉

[1]**East China Normal University**

## Abstract

Reactive dance generation (RDG), the task of generating a dance conditioned on a leader's motion and music, holds significant promise for enhancing human-robot interaction and immersive digital entertainment. Despite progress in duet synchronization and motion-music alignment, two key challenges remain: generating fine-grained spatial interactions and ensuring long-term temporal coherence. In this work, we introduce **ReactDance**, a diffusion framework that operates on a novel hierarchical latent space to address these spatiotemporal challenges in RDG. First, for high-fidelity spatial expression and fine-grained control, we propose Hierarchical Finite Scalar Quantization (**HFSQ**). This multi-scale motion representation effectively disentangles coarse body posture from high-frequency dynamics, enabling independent and detailed control over both aspects through a layered guidance mechanism. Second, to efficiently generate long sequences with high temporal coherence, we propose Blockwise Local Context (**BLC**), a non-autoregressive sampling strategy. Departing from slow, frame-by-frame generation, BLC partitions the sequence into blocks and synthesizes them in parallel via periodic causal masking and positional encodings. Coherence across these blocks is ensured by a dense sliding-window training approach that enriches the representation with local temporal context. Extensive experiments show that ReactDance substantially outperforms state-of-the-art methods in motion quality, long-term coherence, and sampling efficiency. Project page: https://ripemangobox.github.io/ReactDance.

## 1 Introduction

Reactive Dance Generation (RDG) aims to synthesize lifelike and artistically coherent reactor motions that align with a lead dancer and accompanying music. This form of responsive interaction embodies a complex interplay of artistic expression and interpersonal dynamics that is a key challenge in fields like social robotics and human-agent interaction. The ability to generate such intricate, reactive behaviors holds significant potential for applications like virtual avatar animation for gaming and the metaverse (Hu, 2024; Zhang et al., 2023), as well as embodied AI and human-robot interaction (Granados et al., 2017; Chen et al., 2025).

Despite its promise, RDG faces two critical, unaddressed challenges: (1) modeling fine-grained spatial interactions and (2) maintaining long-term temporal coherence. While recent methods have advanced duet synchronization using tailored network architectures (Yao et al., 2023; Liang et al., 2024; Ghosh et al., 2024), physical constraints (Wang et al., 2024; Xu et al., 2024), or reinforcement learning (Siyao et al., 2024), they fail to resolve these core issues and fall short of generating truly authentic dance sequences. Their reliance on holistic, high-level constraints overlooks the subtle yet decisive local movements (*e.g.*, the whip-like *boleo* in Tango), producing movements that are synchronized yet artistically sterile. Moreover, a fundamental difficulty for RDG is ensuring long-term temporal coherence. Models are typically trained on short clips, a practice driven by both the

---
✉ Corresponding Author.

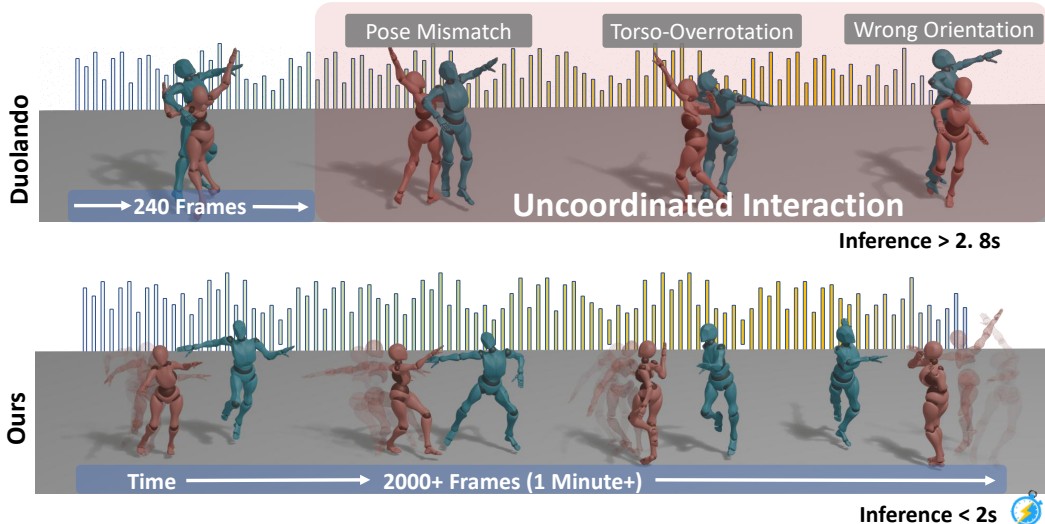

Figure 1: **ReactDance** generates high-fidelity, long-form reactive dance Conditioned on a leader and music, our model uses a hierarchical representation to refine motion from coarse (translucent) to fine-grained (solid). Its parallel sampling mechanism produces coherent sequences exceeding 2000 frames within 2 seconds, avoiding the quality degradation and slow speed of autoregressive sampling.

algorithmic complexity of learning long-term dependencies and practical computational constraints. This creates an inherent discrepancy between training and inference, leading to error accumulation that manifests as temporal drift and an eventual breakdown of synchronization.

To address these limitations, we draw inspiration from structural principles of dance theory and propose two core concepts for generating high-fidelity reactive motion:(1) *Hierarchical Motion Composition:* The principle that dance performance is inherently layered, with foundational full-body rhythms providing a coarse structure and fine-grained dynamics adding detailed semantics (Seham, 2016).(2) *Modular Temporal Coherence:* The observation that long-form choreography achieves consistency by composing shorter, coherent motion phrases and ensuring smooth transitions between them (Krug, 2022).

Grounded in these principles, we propose ReactDance, a two-stage diffusion framework that operationalizes this hierarchical and modular approach.

First, to achieve hierarchical refinement, we introduce a Hierarchical Finite Scalar Quantization (HFSQ) representation. Unlike traditional VQ-VAE models (Gong et al., 2023; Jiang et al., 2024) prone to codebook collapse, HFSQ combines the stability of FSQ (Mentzer et al., 2024) with a hierarchical structure (Yang et al., 2023) to build a robust, multi-scale latent space. This representation explicitly disentangles coarse global movements from fine-grained local details. Building on this, we develop Layer-Decoupled Classifier-Free Guidance (LDCFG), a technique providing independent, layered control over motion semantics at different granularities.

Second, to ensure long-term coherence, we introduce the Blockwise Local Context (BLC) sampling mechanism. BLC replaces fragile autoregressive sampling with efficient, parallel generation of all motion blocks. Long-term consistency is achieved by precisely aligning each block's temporal context at inference with the windows learned during our dense sliding-window (DSW) training. This alignment, enforced using periodic causal masking and positional encodings, effectively eliminates temporal drift. As a result, ReactDance efficiently generates coherent sequences exceeding 60 seconds within 2 seconds.

In summary, our main contributions are:

- We introduce ReactDance, a hierarchical framework for Reactive Dance Generation (RDG) that produces high-fidelity, coherent reactive dance sequences. It leverages a novel Hierarchical Finite Scalar Quantization (HFSQ) representation to progressively model the reactive performance from

coarse body rhythms to fine-grained local movements, effectively addressing challenges in spatial interaction refinement.

- We propose Blockwise Local Context (BLC), a novel parallel sampling mechanism that enables coherent generation of long motion sequences within 2 seconds for over 2000 frames, building on a temporally continuous latent space. By decomposing the generation process into blocks aligned with the temporal context of training, BLC mitigates error accumulation inherent in prior autoregressive methods, achieving state-of-the-art long-term stability and sampling efficiency.

- We propose Layer-Decoupled Classifier-Free Guidance (LDCFG), a novel guidance technique tailored for the HFSQ representation. LDCFG enables independent conditioning over different scales of the generated motion, offering finer-grained semantic manipulation of reactive performances compared to existing approaches.

## 2 RELATED WORKS

### 2.1 MUSIC-DRIVEN DANCE GENERATION

Music-driven dance generation has advanced from solo to multi-dancer interactions. Early music-driven dance generation relied on sequence-to-sequence models like LSTMs (Tang et al., 2018) and GANs (Li et al., 2022), but often suffered from rigid transitions and a lack of detail. While auto-regressive models (Li et al., 2021; Siyao et al., 2022) improved solo dance quality, they lacked mechanisms for duet interactions. Recent duet-specific models still fall short; approaches like DanY (Yao et al., 2023) and Duolando (Siyao et al., 2024) use single-scale representations, causing synchronization artifacts, while DuetGen's (Ghosh et al., 2025) coupled representation models the joint distribution, inherently limiting the conditional diversity of reactor responses given a fixed leader. In contrast, we introduce a hierarchical representation that captures multi-scale spatial dynamics to generate high-fidelity, coherent reactive duets.

### 2.2 HIERARCHICAL MOTION GENERATION MODELS

The quality and controllability of generated motion are critically dependent on its underlying representation. Codebook-based models (Gong et al., 2023; Jiang et al., 2024) are prone to codebook collapse and detail loss from VQ-VAE (Oord et al., 2017) quantization. Although MoMask (Guo et al., 2024) introduces residual structures, its reliance on RVQ-VAE and serial generation causes cumulative errors across scales. While diffusion models (Qi et al., 2025; Liu et al., 2025) achieve state-of-the-art fidelity, they typically operate on flat representations, lacking a structure for multi-level control. Existing hierarchical models (Qi et al., 2023; Li et al., 2024) have focused on temporal, not spatial, refinement. Our work fills this gap by introducing a spatial hierarchy that enables both high-fidelity synthesis and multi-scale control over the generated motion.

### 2.3 LONG MOTION SEQUENCE GENERATION

Generating coherent long motion sequences remains challenging. For text-to-motion, methods like MotionStreamer (Xiao et al., 2025) and priorMDM (Shafir et al., 2024a) use two-pass refinement to ensure smooth transitions between clips. The problem is harder in music-to-dance, which requires long-range musical coherence (Tsakalidis, 2020). Prior solutions are often inefficient, such as the iterative inpainting in EDGE (Tseng et al., 2023), or inflexible, like the primitive-based temporal hierarchy in Lodge (Li et al., 2024). In contrast, our approach enables efficient, parallel generation by training on overlapping windows, directly embedding temporal context into our representation and avoiding the costly overhead of previous methods.

## 3 METHOD

Our proposed framework, ReactDance, generates a coherent long reactive dance sequence in a two-stage process. First, an autoencoder with a Hierarchical Finite Scalar Quantizer (HFSQ) bottleneck learns to encode dance movements into a hierarchical latent representation. Second, a conditional diffusion model learns to generate HFSQ latent representations based on leader movements and background music, which are then decoded into the final motion.

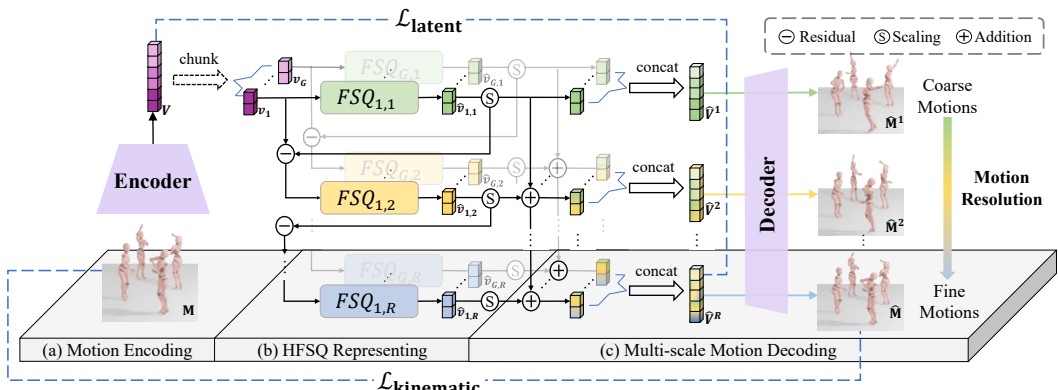

Figure 2: **Motion HFSQ overview.** The proposed motion HFSQ learns to progressively encode motion sequence into a hierarchical representation $\mathcal{V} = \left\{ \hat{\boldsymbol{v}}_{g,r} \right\}_{g=1, r=1}^{G, R}$ and reconstructs motions via a grouped residual architecture. Building on FSQ, HFSQ eliminates codebook collapse while enhancing multi-scale motion expressiveness through grouped residual quantization, which integrates coarse-to-fine motion semantics.

## 3.1 MOTION AND MUSIC REPRESENTATION

**Motion Representation.** We represent a dance sequence of $N$ frames using a skeleton with $J = 22$ joints from the SMPL model (Loper et al., 2023). To better capture the dynamics of reactive dance, our approach utilizes an asymmetric representation for the leader and the reactor.

The leader's motion is modeled holistically as a single sequence of global joint positions, denoted as $\mathbf{M}_L \in \mathbb{R}^{N \times J \times 3}$. In contrast, the reactor's motion is decomposed into three independent components: (1) Upper-body motion ($\mathbf{M}_{Rup} \in \mathbb{R}^{N \times J_{up} \times 3}$): The localized positions of the $J_{up}$ upper-body joints relative to the reactor's own root joint. (2) Lower-body motion ($\mathbf{M}_{Rdown} \in \mathbb{R}^{N \times J_{down} \times 3}$): The localized positions of the $J_{down}$ lower-body joints relative to the reactor's own root joint. (3) Relative root distance ($\mathbf{M}_{tr} \in \mathbb{R}^{N \times 3}$): The displacement vector between the leader's and reactor's roots.

**Music Representation.** For the accompanying music, we use Librosa (Brian McFee et al., 2015) to extract a rich acoustic feature vector $\boldsymbol{c} \in \mathbb{R}^{54}$ for each corresponding motion frame. This vector comprises 20-dimensional MFCCs, their 20-dimensional deltas, 12-dimensional chroma features, and 1-dimensional onset strength and beat tracking signals.

Our goal is thus to generate the set of reactor motion components $\mathbf{M}_R = \{\mathbf{M}_{Rup}, \mathbf{M}_{Rdown}, \mathbf{M}_{tr}\}$ conditioned on the music $\boldsymbol{c}$ and the leader's motion $\mathbf{M}_L$. To process the complete reactor motion $\mathbf{M}_R$, each of its components is passed through a dedicated and structurally identical HFSQ stream. For clarity, the following sections will describe the core methodology for a single holistic stream.

## 3.2 HIERARCHICAL MOTION REPRESENTATION VIA HFSQ

We learn a compressed and structured representation of motion using a hierarchical finite scalar quantizer, inspired by recent work in neural audio codecs (Yang et al., 2023; Mentzer et al., 2024).

**Encoder and Quantizer.** A temporal encoder, consisting of 1D convolutional layers, maps an input reactor motion sequence $\mathbf{M}_R \in \mathbb{R}^{N \times J_R}$ to a downsampled feature sequence $\boldsymbol{v} \in \mathbb{R}^{n \times d}$, where $n = N/4$. This sequence is then processed by our HFSQ module, which operates in two stages:

- **Grouping:** The feature vector $\boldsymbol{v}$ is split into $G$ parallel groups, $\boldsymbol{v} = [\boldsymbol{v}_1, \ldots, \boldsymbol{v}_G]$, where each group $\boldsymbol{v}_g \in \mathbb{R}^{n \times (d/G)}$.

- **Cascaded Residual Quantization:** Each group $\boldsymbol{v}_g$ is quantized sequentially over $R$ residual stages. Let the initial residual be $\boldsymbol{e}_{g,1} = \boldsymbol{v}_g$. For each stage $r \in [1, \ldots, R]$, we compute:

$$\boldsymbol{z}_{g,r} = \text{FSQ}_r(\boldsymbol{e}_{g,r}); \quad \hat{\boldsymbol{v}}_{g,r} = \text{Dequantize}(\boldsymbol{z}_{g,r}); \quad \boldsymbol{e}_{g,r+1} = \boldsymbol{e}_{g,r} - s_r \cdot \hat{\boldsymbol{v}}_{g,r}. \tag{1}$$

The core output is the set of continuous reconstructions $\hat{\boldsymbol{v}}_{g,r}$ from each hierarchical level. We define the complete hierarchical continuous representation as the set of these vectors, denoted by $\mathcal{V}$:

$$\mathcal{V} = \{\hat{\boldsymbol{v}}_{g,r}\}_{g=1,r=1}^{G,R}. \tag{2}$$

**Frequency-based Semantic Decomposition.** Crucially, this residual hierarchy naturally disentangles motion semantics based on signal energy. The base layer ($r = 1$), minimizing the primary reconstruction error, captures the **Coarse Motion** (*e.g.*, global posture, orientation, and low-frequency trajectories). Subsequent layers ($r \geq 2$), encoding the residual error, capture the **Fine Motion** (*e.g.*, high-frequency local dynamics and subtle articulation details). As we will detail, this structured set $\mathcal{V}$ serves as the target space for our diffusion model. Notably, we find that a lightweight configuration with just two residual quantization layers ($R = 2$) is sufficient for our final architecture, with more details available in Appendix A.6.

**Decoder and Training Objective.** The decoder reconstructs the motion from the dequantized features. The final feature vector $\hat{\boldsymbol{v}}$ is obtained by summing the reconstructions within each group and concatenating them: $\hat{\boldsymbol{v}}_g = \sum_{r=1}^{R} s_r \cdot \hat{\boldsymbol{v}}_{g,r}$ and $\hat{\boldsymbol{v}} = [\hat{\boldsymbol{v}}_1, \dots, \hat{\boldsymbol{v}}_G]$. The HFSQ autoencoder is trained to minimize a combination of a latent loss and physical plausibility losses on the final decoded motion $\hat{\mathbf{M}}_R = \text{Dec}(\hat{\boldsymbol{v}})$:

$$\mathcal{L}_{\text{HFSQ}} = \lambda_{kin}\mathcal{L}_{\text{kinematic}} + \lambda_{lat}\mathcal{L}_{latent}, \quad \text{where} \tag{3}$$

$$\mathcal{L}_{\text{kinematic}} = \sum_{k \in \{\text{pos,vel,acc}\}} \lambda_{Pk} \cdot \|\mathbf{M}_R^{(k)} - \hat{\mathbf{M}}_R^{(k)}\|_1, \tag{4}$$

$$\mathcal{L}_{latent} = \|\boldsymbol{v} - \text{sg}(\hat{\boldsymbol{v}})\|_2^2 + \beta\|\text{sg}(\boldsymbol{v}) - \hat{\boldsymbol{v}}\|_2^2. \tag{5}$$

The $\mathbf{M}^{(k)}$ denotes the $k$-th order temporal derivative (position, velocity, acceleration) of the ground-truth motion, and $\hat{\mathbf{M}}^{(k)}$ denotes the reconstructed counterparts. This objective ensures that the final reconstruction is both physically realistic and faithful to the encoder's output.

**Latent Robustness via Progressive Masking.**

To construct a structured HFSQ latent space, we corrupt the hierarchical representation $\mathcal{V}$ (Eq. 2) using two complementary techniques. First, residual masking promotes inter-layer independence by stochastically occluding higher-level FSQ layers while perturbing the base layer with scaled Gaussian noise. Second, code masking improves the decoder's robustness by randomly masking dimensions within individual FSQ codes. Together, these operations regularize the latent space, improving the decoder's tolerance to imperfect inputs and thereby enhancing the final motion quality.

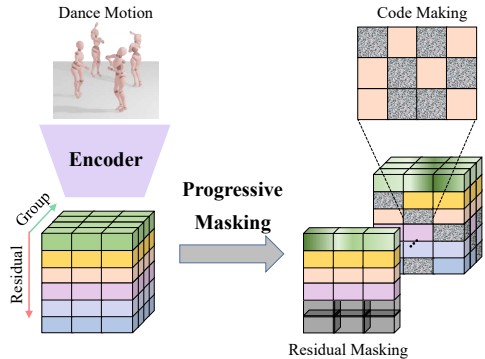

Figure 3: **Progressive masking overview.**

### 3.3 LATENT DIFFUSION ON HIERARCHICAL FEATURES

**Generative Target.** Unlike methods that diffuse on discrete codes or the final aggregated feature, our diffusion model operates on the hierarchical continuous representation $\mathcal{V}$ (Eq. 2). This strategy enables the model to learn the coarse-to-fine structure of motion within the continuous space defined by our HFSQ module. The final motion is then robustly decoded from this generated representation.

**Diffusion Framework.** We model the generation process using a DDPM (Ho et al., 2020). The forward process gradually adds Gaussian noise to the HFSQ latents $\boldsymbol{x}_0 = \mathcal{V}$ over T timesteps. The reverse process trains a Transformer-based network $\mathcal{G}_\theta$ to denoise the corrupted latents $\boldsymbol{x}_t$ at timestep $t$, conditioned on music c and leader motion $\mathbf{M}_L$. To leverage the hierarchical structure of $\mathcal{V}$, we apply independent weights to the loss for each residual scale. The latent diffusion loss is:

$$\mathcal{L}_{\text{simple}} = \sum_{r=1}^{R} \lambda^r \cdot \mathbb{E}_{\boldsymbol{x}_0,t,\epsilon} \left[ \|\mathcal{V}^r - \mathcal{G}_\theta(\boldsymbol{x}_t, t, \boldsymbol{c}, \mathbf{M}_L)^r\|_2^2 \right], \tag{6}$$

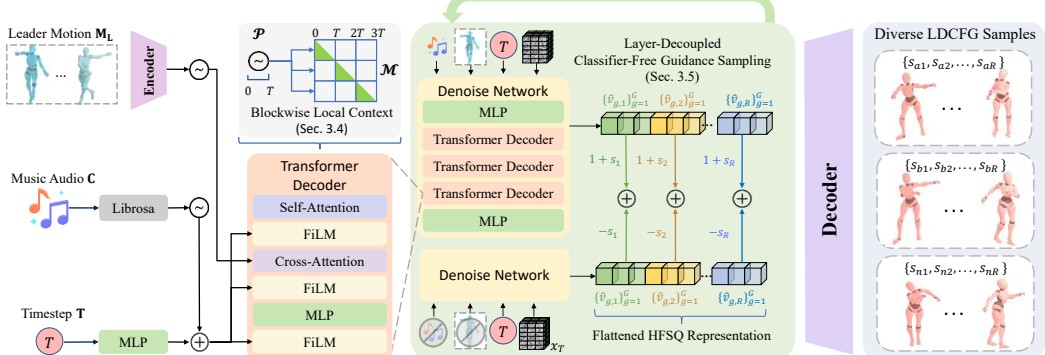

Figure 4: **ReactDance Pipeline Overview.** Our ReactDance generates long, high-fidelity reactive dance sequences conditioned on leader motion and music. The core is a diffusion model that learns to denoise hierarchical HFSQ latents. Leader motion is injected via cross-attention, while music features are fused using a FiLM layer. For coherent generation of long sequences, our Blockwise Local Context (BLC) sampling strategy partitions the timeline into parallel blocks with aligned temporal contexts. Within each denoising step, Layer-Decoupled Classifier-Free Guidance (LDCFG) provides fine-grained control by applying independent guidance weights to each HFSQ scale. Finally, the denoised latents are decoded into a dance sequence coherently aligned with the input conditions.

where $\mathcal{V}^r = \{\hat{v}_{g,r}\}_{g=1}^G$ represents the set of feature vectors at the $r$-th level, and $\mathcal{G}_\theta(\cdot)^r$ is the corresponding portion of the network's output. The weights $\lambda^r$ balance the contribution of each residual level, allowing the model to focus on coarse-to-fine details during generation. Moreover, we apply kinematic loss $\mathcal{L}_{kinematic}$ (Eq. 4), foot contact loss $\mathcal{L}_{fc}$, leader-reactor relative orientation loss $\mathcal{L}_{ro}$ (Liang et al., 2024) to enforce physical plausibility and interaction coherence. The complete loss is defined in Eq. 7, with $\lambda_{\text{kin}}$, $\lambda_{\text{fc}}$ and $\lambda_{\text{ro}}$ as balancing weights:

$$\mathcal{L}_{\text{diffusion}} = \mathcal{L}_{\text{simple}} + \lambda_{\text{kin}}\mathcal{L}_{\text{kinematic}} + \lambda_{\text{fc}}\mathcal{L}_{\text{fc}} + \lambda_{\text{ro}}\mathcal{L}_{\text{ro}}. \tag{7}$$

## 3.4 EFFICIENT LONG SEQUENCE GENERATION VIA BLOCKWISE LOCAL CONTEXT

Generating long, coherent dance sequences is a critical challenge, as the desired inference length $K$ often far exceeds the model's training window $T$. While autoregressive methods can extend sequence length, they suffer from serial inference latency and error accumulation. To overcome these limitations, we introduce **Blockwise Local Context (BLC)**, a non-autoregressive sampling strategy that generates partitioned long sequences in parallel. BLC achieves high-fidelity generation through two synergistic principles: *Intra-block Consistency* (via sampling protocols) and *Inter-block Continuity* (via training dynamics).

**Intra-block Consistency via Parallel Sampling.** To enable parallel generation, BLC aligns the inference context of an arbitrary long sequence with the fixed-length context seen during training. We achieve this through:

- **Periodic Causal Masking:** A block-diagonal attention mask $\mathcal{M}$ is applied across the sequence to enforce causality strictly within non-overlapping blocks of size $T$, preventing information leakage and halting error propagation between blocks.

- **Phase-aligned Positional Encoding:** A periodic positional encoding $\mathcal{P}_i$ is introduced to reset the temporal phase for each block:

$$\mathcal{P}_i = \sin\left(\frac{\pi(i \bmod T)}{T}\right) \oplus \cos\left(\frac{\pi(i \bmod T)}{T}\right), \tag{8}$$

where $i$ is the global temporal position. This ensures that every inference block acts as an independent, well-formed window identical to the training distribution.

**Inter-block Continuity via Dense Sliding Window (DSW).** A core challenge in parallel block sampling is ensuring seamless transitions at the boundaries between independently generated blocks.

We resolve this not by explicit post-processing, but by enforcing a Latent Manifold Constraint during training via our Dense Sliding Window (DSW) strategy. We train the model with a stride $m$ significantly smaller than the window size ($m \ll T$, *e.g.*, $m = 4, T = 240$). This design is crucial for two reasons:

- **Phase-Agnostic Transition Prior:** By processing motion windows at every possible phase shift, the model observes any given motion frame $f_t$ in diverse temporal roles (as a start, middle, or end frame). Consequently, the decoder learns a **phase-agnostic transition function**, understanding that adjacent latents $z_t$ and $z_{t+1}$ must resolve to continuous motion regardless of where the window boundary cuts.
- **Decoder as Kinematic Smoother:** During inference, even if the parallel attention blocks create slight boundary discontinuities in the latent space, the HFSQ decoder—having been trained on thousands of overlapping, phase-shifted windows—projects these boundary latents onto the valid continuous motion manifold. This implicitly stitches blocks $i$ and $i + 1$ with smooth, physically plausible transitions.

In summary, BLC's masking enables efficiency through parallelism, while DSW's dense supervision ensures coherence by learning robust boundary invariance.

### 3.5 LAYER-DECOUPLED CLASSIFIER-FREE GUIDANCE

**Precision Control vs. Spatial Control.** While our Multi-stream Input design (Section 3.1) explicitly handles *Spatial Control* by defining specific body regions for interaction, ReactDance requires a complementary mechanism for *Precision Control*—allowing users to modulate *how* strictly the generation adheres to conditioning signals across different semantic scales. Standard Classifier-Free Guidance (CFG) (Ho & Salimans, 2021) applies a single scalar weight across the entire motion representation (Tevet et al., 2023; Shafir et al., 2024b), forcing a coupled trade-off between global pose stability and fine-grained interaction details. This is suboptimal for hierarchical tasks where structural coherence and subtle dynamics may require different guidance strengths.

To address this, we propose **Layer-Decoupled Classifier-Free Guidance (LDCFG)**. This method leverages the hierarchical nature of HFSQ for orthogonal control over motion structure and details.

**Training Strategy: Independent Condition Dropout.** To support decoupled guidance during inference, the model must learn to condition each hierarchical layer independently. Unlike standard CFG training which drops conditions for all layers simultaneously, we employ *Independent Condition Dropout*. During training, the conditioning signal ($c$ and $\mathbf{M}_L$) for the $r$-th residual layer is replaced with a null embedding $\emptyset$ with an independent probability $p = 0.2$. This forces the denoiser to learn the conditional distribution $P(z^r|c, \mathbf{M}_L)$ marginally, laying the foundation for layer-isolated control during inference.

**Inference Mechanism.** During sampling, LDCFG assigns an independent guidance strength $s_r$ to each scale of the HFSQ representation. The predicted noise is computed as:

$$\hat{x}_0^r = (1 + s_r)\mathcal{G}_\theta(x_t^r, t, c, \mathbf{M}_L) - s_r\mathcal{G}_\theta(x_t^r, t, \emptyset, \emptyset), \tag{9}$$

where $s_r$ controls the guidance scale for the $r$-th layer. This formulation provides an explicit mechanism for frequency-based control by modulating the coarse and fine components defined in Section 3.2. The **Base Guidance ($s_1$):** Modulates the *Coarse Motion*. Increasing $s_1$ anchors the global posture and orientation, ensuring kinematic stability. While the **Residual Guidance ($s_{r \geq 2}$):** Modulates the *Fine Motion*. Increasing $s_2$ specifically enhances the subtle local dynamics and interaction nuances without altering the macro-level motion structure.

This disentanglement allows ReactDance to act as a "Detail Specialist" or "Structure Specialist" as needed, achieving the optimal balance between fidelity and diversity.

## 4 EXPERIMENT

### 4.1 DATASETS AND EXPERIMENTAL SETTING

**Dataset.** We conduct experiments on the DD100 dataset (Siyao et al., 2024), which comprises 1.95 hours of paired dance motion and music data spanning 10 musical genres. We adopt the original

Table 1: Comparison with state-of-the-art methods on the test dataset of DD100 dataset. Symbols ↑, ↓, and → indicate the higher, lower and closer to Ground Truth are better. **Bold** and underline indicate the best and second best results. The dotted line separates methods for single-person motion generation (above) from those for duet and multi-person generation (below).

| Method | Solo Metrics | | | | | | | | Interactive Metrics | | | | |
| | $FID_k(\downarrow)$ | $FID_g(\downarrow)$ | $Div_k(\rightarrow)$ | $Div_g(\rightarrow)$ | $MPJPE(\downarrow)$ | $MPJVE(\downarrow)$ | $PFC(\downarrow)$ | $BAS(\rightarrow)$ | $FID_{cd}(\downarrow)$ | $Div_{cd}(\rightarrow)$ | $BED(\rightarrow)$ | $IPR(\downarrow)$ | $AITS(\downarrow)$ |
|---|---|---|---|---|---|---|---|---|---|---|---|---|---|
| Ground Truth | - | - | 10.86 | 7.82 | - | - | - | 0.1791 | - | 12.53 | 0.5308 | - | - |
| GestureLSM | 14.65 | 34.23 | 12.20 | 9.00 | 171.37 | 19.01 | 0.7903 | **0.1734** | 40.09 | 9.45 | 0.1903 | 19.01 | 3.13 |
| EDGE | 68.68 | 113.25 | 5.62 | 10.05 | 235.52 | 20.76 | 0.6226 | 0.1854 | 1523.27 | **12.37** | 0.1904 | 8.44 | 2.91 |
| TCDiff | 105.97 | 251.35 | 14.23 | 24.54 | 182.64 | 22.91 | 1.1165 | 0.2270 | 1472.00 | 11.31 | 0.2304 | **7.58** | 3.34 |
| InterGen | 35.89 | 47.28 | 12.24 | 9.13 | 210.66 | 18.69 | 0.9842 | 0.1634 | 176.19 | 18.10 | 0.2746 | 17.58 | 5.22 |
| Duolando | 27.68 | 35.01 | 10.95 | 8.70 | 174.54 | 18.72 | 0.9276 | 0.2086 | 17.49 | 14.73 | 0.3285 | 17.42 | 4.41 |
| **ReactDance** | **5.57** | **7.63** | **10.82** | **7.76** | **132.99** | **15.68** | **0.6039** | 0.2031 | **14.17** | 10.58 | **0.3863** | 7.84 | **1.75** |

dataset's training/test splits. All motion sequences are split into 8-second clips (30 frames per second) with a sliding window stride of 4.

**Evaluation Metrics.** We adopt a comprehensive set of quantitative metrics to evaluate the generated motion from four key perspectives: (1) Motion Quality. To assess the realism of individual motions, we measure distributional similarity and reconstruction accuracy. Distributional metrics include Fréchet Inception Distance (FID) and Diversity (Div) on both kinematic ($FID_k$, $Div_k$) (Onuma et al., 2008) and graphical ($FID_g$, $Div_g$) (Müller et al., 2005) features. Reconstruction accuracy against the ground truth is measured by Mean Per-Joint Position Error (MPJPE) for static poses and Mean Per-Joint Velocity Error (MPJVE) for dynamics (Ghosh et al., 2024). Additionally, the Physical Foot Contact (PFC) (Tseng et al., 2023) detects foot skating artifacts by verifying kinematic consistency. (2) Interaction Quality. To evaluate leader-reactor coordination, we compute FID and Diversity on cross-distance features ($FID_{cd}$, $Div_{cd}$), which describe the evolving spatial relationships between the dancers' key joints. To assess physical validity, we calculate the Inter-Penetration Rate (IPR), quantifying vertex-mesh penetrations using the Generalized Winding Number field (Jacobson et al., 2013), accelerated by AABB-based culling. (3) Music-Dance Alignment. We use two metrics: the Beat Align Score (BAS) (Siyao et al., 2022) to measure synchronization between motion peaks and music beats, and the Beat Echo Degree (BED) (Siyao et al., 2024) to quantify rhythmic consistency between the dancers. (4) Generation Efficiency. We report the Average Inference Time per Sequence (AITS) in seconds.

## 4.2 Comparison with Baseline Methods

**Baselines.** We compare our model against several state-of-the-art (SOTA) methods. For a fair comparison, all baselines were adapted for the reactive dance generation (RDG) task and retrained from scratch on the DD100 dataset using the SMPL body model. We integrated the leader's motion as a direct condition for several models: the GestureLSM (Liu et al., 2025) was modified by replacing its text encoder with a leader motion encoder, while for the music-driven EDGE (Tseng et al., 2023), we added the features from a leader motion encoder to its existing music features. Other baselines required more targeted modifications. The text-to-interaction model InterGen (Liang et al., 2024) was adapted by replacing its text encoder with a music encoder and changing its weight-shared network to be independent for each dancer. For the group dance model TCDiff (Dai et al., 2025), we altered its navigator module to predict the reactor's trajectory from the leader's motion. Finally, Duolando (Siyao et al., 2024), an existing RDG model, was retrained on our dataset to ensure body model consistency (SMPL vs. its native SMPL-X).

**Analysis.** As reported in Table 1, our method, ReactDance, demonstrates superior performance across the majority of metrics. Note that these results are computed on the entire test set, which features long sequences with an average length of 2066 frames. This highlights our model's strong generalization capability and consistency in long-term generation. The significant lead in realism ($FID_k$, $FID_g$) and accuracy (MPJPE, MPJVE) stems from our effective HFSQ representation, which enables the diffusion model to learn motion hierarchically from coarse postural dynamics to fine-grained articulations. This fine-grained capability also ensures high physical plausibility, as evidenced by the lowest PFC score (0.6039), indicating minimized foot skating artifacts compared to baselines. Regarding duet coherence, ReactDance significantly outperforms all baselines, achieving a $FID_{cd}$

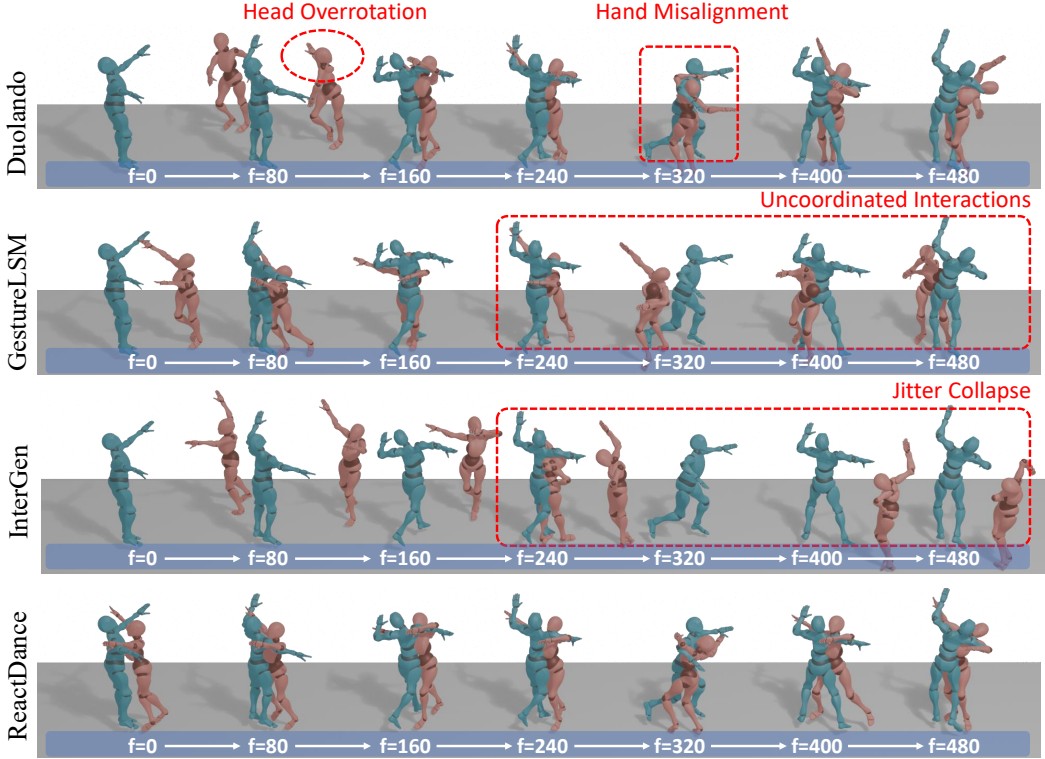

Figure 5: **Qualitative comparison of reactive dance generation.** Given the same leader motion, Duolando produces unnatural head rotations and GestureLSM shows uncoordinated interactions. InterGen's motion collapses into unrealistic jitter when generalizing beyond its training horizon. In contrast, our model generates a fluid and coherent reactive motion.

of 14.17 and a BED of 0.3863. Crucially, our method maintains a low IPR (7.84%), significantly surpassing the autoregressive baseline Duolando (17.42%) and proving that our model generates spatially coherent interactions without severe collision artifacts. Furthermore, ReactDance also achieves the most efficient generation with an AITS of 1.75s due to the parallel generation capability of our BLC sampling strategy. In contrast, competing methods exhibit key limitations. Architectures with a two-stage design like GestureLSM and Duolando, while competitive in solo motion quality, are constrained by their representations. GestureLSM's use of a single-scale latent space for generation limits its ability to capture hierarchical interactions (evidenced by a high $FID_{cd}$ of 40.09), while Duolando's VQ-VAE is less effective at fine-grained reconstruction. Other models like EDGE and TCDiff perform poorly overall, as they learn directly on the high-frequency raw motion space and lack explicit mechanisms for interaction modeling.

**User Study.** We conducted a user study to evaluate the perceptual quality of long-form ($> 60$ seconds) reactive dance sequences, involving 20 participants. Each participant completed 15 paired comparisons, rating videos generated by our method against baseline methods on a 5-point Likert scale across three criteria: Motion Naturalness, Music Alignment, and Interaction Coordination. As detailed in the appendix (Fig. 9), our method consistently outperformed all baselines across each criterion, demonstrating a strong and consistent participant preference. These results highlight our method's capability to produce high-quality, coherent interactions over extended durations.

## 4.3    ABLATION STUDY

**Effects of HFSQ and Progressive Masking.**

We conduct a comprehensive ablation study on the tokenizer architecture and Progressive Masking (PM) training strategy in Table 2. Comparing different architectures, the VAE suffer from posterior collapse, leading to over-smoothed reconstruction (MPJPE 40.87) and poor generation alignment

(MPJPE 230.21). Standard RVQ-VAE with same number of residual layers ($R = 2$) improves upon VAE but still lags behind HFSQ, particularly in generation quality ($\text{FID}_g$ 26.98 vs. 7.63), even when trained with PM strategy. We attribute this to the *irregular topology* of vector codebooks, where the latent of adjacent indices lack semantic continuity, creating a difficult optimization landscape for diffusion. In contrast, HFSQ projects motion onto a fixed scalar grid that preserves *ordinal relations*, providing a smoother, "diffusion-friendly" manifold. Finally, analyzing the PM strategy reveals a critical trade-off: removing the PM strategy trades a marginal MPJPE improvement for a substantial drop in realism ($\text{FID}_g$ increases to 10.46), demonstrating that PM is crucial for synthesizing high-fidelity motion. Additional results are available in the Appendix A.3.

Table 2: Ablation study of HFSQ tokenizer and the Progressive Masking (PM).

| Method | Solo Metrics | | | |
| --- | --- | --- | --- | --- |
| | $\text{FID}_g(\downarrow)$ | $\text{Div}_g(\rightarrow)$ | MPJPE($\downarrow$) | BAS($\rightarrow$) |
| Ground Truth | – | 7.82 | – | 0.1791 |
| *Stage 1: Reconstruction* | | | | |
| VAE | 5.32 | **7.78** | 40.87 | 0.1805 |
| RVQ-VAE (w/o. PM) | 4.57 | 7.72 | 36.59 | 0.1841 |
| RVQ-VAE (w PM) | 4.09 | 7.73 | 32.98 | 0.1842 |
| HFSQ (w/o. PM) | 3.77 | 7.74 | **24.15** | **0.1784** |
| HFSQ (w PM, Ours) | **3.56** | 7.73 | 28.66 | 0.1816 |
| *Stage 2: Generation* | | | | |
| VAE | 18.99 | 7.77 | 230.21 | 0.2705 |
| RVQ-VAE (w/o. PM) | 36.73 | 8.82 | 139.42 | 0.2040 |
| RVQ-VAE (w PM) | 26.98 | 7.75 | 138.28 | **0.1946** |
| HFSQ (w/o. PM) | 10.46 | 7.65 | **132.19** | 0.2003 |
| **Ours (HFSQ w PM)** | **7.63** | 7.76 | 132.99 | 0.2031 |

**Effects of Blockwise Local Context Sampling.** We ablate the two core components of our Blockwise Local Context (BLC) sampling to validate its contributions to duet coherence and sampling efficiency, with results presented in Table 3. We analyze the dense sliding window component by increasing the training stride from 4 (our full model) to 16 and 64. We observe a significant degradation in coherence, as evidenced by the substantial increase in $\text{FID}_{cd}$ and the decline in BED. Moreover, we replace our periodic positional encoding (PPE) and periodic causal attention masking (PCAM) of BLC with the latent stitching operation (Tseng et al., 2023). This change harms both sampling efficiency (AITS increases from 1.75 to 2.03) and generation quality ($\text{FID}_{cd}$ increases from 14.17 to 18.59). These results confirm a dense training stride is crucial for coherence, and our periodic sampling mechanism is key to achieving both high quality and efficiency. For a comprehensive analysis of the trade-off between motion continuity and training cost, please refer to Appendix A.7.

Table 3: Ablation of Dense Sliding Window (DSW) training and BLC sampling strategies.

| Method | Duet Metrics | | | |
| --- | --- | --- | --- | --- |
| | $\text{FID}_{cd}(\downarrow)$ | $\text{Div}_{cd}(\rightarrow)$ | BED($\rightarrow$) | AITS($\downarrow$) |
| Ground Truth | – | 12.53 | 0.5308 | – |
| s=64 | 39.50 | 12.08 | 0.2840 | – |
| s=16 | 22.69 | 10.06 | 0.3065 | – |
| w/o. PPE, PCAM | 18.59 | 11.17 | 0.3218 | 2.03 |
| **ReactDance (s=4)** | **14.17** | 10.58 | **0.3863** | **1.75** |

**Effect of Layer-Decoupled Guidance.** Our diffusion model operates on hierarchical HFSQ latents, enabling us to apply independent classifier-free guidance weights $s_r$ of Eq. (9) to each motion scale. This provides fine-grained control over the fidelity-diversity trade-off across different semantic levels. For instance, a user can enforce strong postural alignment with high guidance on coarse scales while encouraging creative local articulation with lower guidance on fine scales. A detailed analysis demonstrating this flexibility is available in Appendix A.4.

## 5 CONCLUSION

We present ReactDance, a diffusion framework for generating high-fidelity and coherent long-form reactive dance. Our method is built upon a hierarchical latent representation that captures multi-scale spatiotemporal dynamics. This representation enables two key contributions: an efficient parallel sampling strategy for long sequence generation and a multi-level conditioning mechanism that offers fine-grained artistic control. By solving the fundamental challenge of maintaining kinematic stability over extended durations, ReactDance serves as a robust cornerstone for coherent long-form reactive dance generation. We hope this foundation paves the way for future research to ascend from physical continuity to high-level semantic modeling, enabling the exploration of narrative-driven choreography and emotional storytelling in reactive dance.

**Limitation.** Despite the effectiveness of our HFSQ representation, it lacks explicit semantic meaning across scales, which reduces control interpretability. Additionally, finger motions are omitted mainly due to the noisy hand data of DD100 dataset, which are crucial for nuanced gestures.

**Acknowledgements.** We sincerely acknowledge the anonymous reviewers for their insightful feedback. This article is partially supported by Natural Science Foundation of China under Grants 62472178Fundamental Research Funds for the Central Universities, the Key Technology Research and Development Program Project of the Shanghai Science and Technology Commission under Grants 25511107200the Open Projects Program of State Key Laboratory of Multimodal Artificial Intelligence Systems (No.MAIS2024111).

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

# A APPENDIX

In the appendix, we present:

- Section A.1: The Use of Large Language Models (LLMs).
- Section A.2: Training Hyper-parameters.
- Section A.3: More Results of Ablation on HFSQ Representation and Progressive Masking.
- Section A.4: More Results of Layer-Decoupled Guidance.
- Section A.5: User Study Results.
- Section A.6: More Results on Residual Stages $R$ and Scalability Analysis.
- Section A.7: Additional Analysis on DSW Stride Selection.
- Section A.8: Hand Motion Modeling Analysis.
- Section A.9: Ablation of Conditioning Signals.
- Section A.10: Generalization via Role Swapping.
- Section A.11: Zero-Shot Generalization on Out-of-Distribution Data.

## A.1 THE USE OF LARGE LANGUAGE MODELS

In the preparation of this manuscript and its accompanying code, we utilized Large Language Models (LLMs) as assistive tools. We wish to clarify that the core scientific contributions—including the problem formulation, the design of our proposed method, and the experimental analysis—are entirely the work of the authors. The use of LLMs was confined to the following supporting tasks:

- **Manuscript Preparation:** LLMs were employed for language refinement. This included improving grammatical correctness, rephrasing sentences for better clarity and flow, and ensuring stylistic consistency throughout the paper. All final text was carefully reviewed and edited by the authors to ensure it accurately reflects our research and contributions.

- **Software Development:** LLMs assisted in writing auxiliary and boilerplate code. Specific use cases included generating scripts for data preprocessing and visualization, creating utility functions, and debugging isolated code segments. The core algorithmic implementation of our proposed method was developed exclusively by the authors.

In all instances, the LLM served as a productivity and quality-enhancement tool. The authors maintain full intellectual responsibility and ownership for all conceptual and experimental aspects of this work.

## A.2 TRAINING HYPER-PARAMETERS

Our framework is implemented in PyTorch and trained on a single NVIDIA RTX L40 GPU. The motion HFSQ is trained for 200 epochs using the Lion optimizer (Chen et al., 2023) with initial learning rate $3e^{-5}$ and a stepwise learning rate scheduler decays the rate by 0.998 per epoch. For diffusion training, the denoise network is trained for 1500 epochs using the Adamw optimizer with initial learning rate $2e^{-4}$ and a stepwise learning rate scheduler decays the rate by 0.998 per epoch. Batch size is set to 256 across all stages.

## A.3 MORE RESULTS OF ABLATION ON HFSQ REPRESENTATION AND PROGRESSIVE MASKING

The visualization in Fig. 6 supports the quantitative findings in the main paper. The ablated model using an RVQ-VAE often generates poses that are individually plausible but contextually incorrect within the duet. This leads to clear visual artifacts such as inaccurate relative positioning and distance between the dancers, as well as failures in fine-grained hand-to-hand or hand-to-body contact. This demonstrates that HFSQ's hierarchical structure is crucial for effectively representing the complex spatial dynamics of dance interactions.

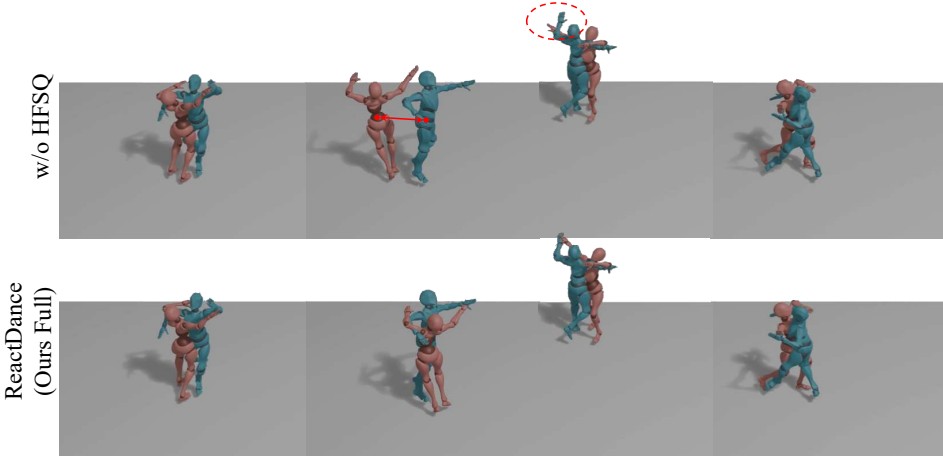

Figure 6: **Qualitative comparison for the HFSQ ablation.** While our full model maintains correct spatial relationships, the model trained without HFSQ (using an RVQ-VAE) fails to capture precise inter-personal dynamics, resulting in incorrect relative distancing and mismatched hand interactions (highlighted in red circles).

As shown in Fig. 7, removing the PM strategy introduces severe interaction artifacts. This aligns with the quantitative trade-off noted in the main paper, where the ablated model achieves a slightly lower MPJPE but a much worse $FID_g$. Without the regularization provided by PM, the model may reconstruct individual poses more closely but fails to learn robust interaction context. This leads to fundamental errors in character orientation relative to the leader and results in physically implausible outcomes, such as body part interpenetration and collisions. This highlights that the PM strategy is essential for learning the physical constraints and relational logic of reactive dance.

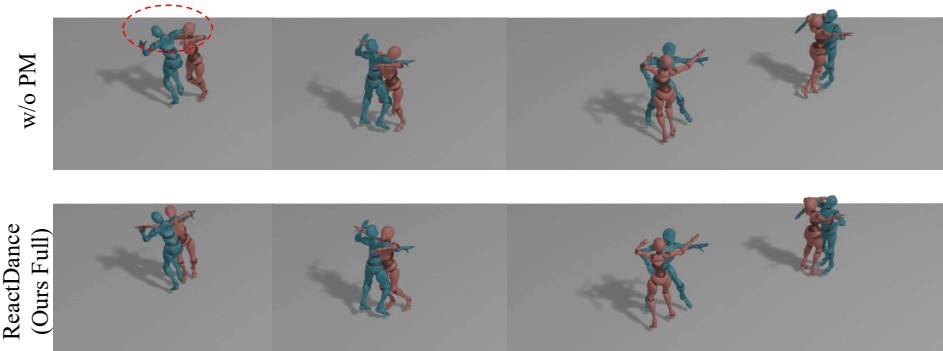

Figure 7: **Qualitative comparison for the Progressive Masking (PM) ablation.** Our full model generates a coherent and physically plausible interaction. In contrast, removing the PM strategy leads to critical errors in interaction logic, such as the reactor adopting an incorrect orientation relative to the leader, resulting in interpenetration artifacts (highlighted).

## A.4 MORE DETAILS ABOUT LAYER-DECOUPLED GUIDANCE

Our Layer-Decoupled Classifier-Free Guidance (LDCFG) is designed to overcome the limitation of a single, global guidance weight used in conventional CFG. By assigning independent guidance strengths $s_r$ (Eq. 9) to each hierarchical layer of our HFSQ representation, LDCFG provides fine-grained control over the fidelity-diversity trade-off at different semantic levels.

We conducted a quantitative analysis by sampling with different LDCFG weight pairs $S = [s_{coarse}, s_{fine}]$, where $s_{coarse}$ and $s_{fine}$ control the coarse- and fine-grained motion scales, respectively. The results are presented in Table 4. The data reveals a clear trade-off between solo motion fidelity

Table 4: Solo and Duet metrics with diverse Layer-Decoupled Classifier-Free Guidance (LDCFG) weights. Best and second-best results are in **bold** and underline, respectively.

| LDCFG Weight | Solo Metrics | | | | | | | Interactive Metrics | | |
| --- | --- | --- | --- | --- | --- | --- | --- | --- | --- | --- |
| | $FID_k(\downarrow)$ | $FID_g(\downarrow)$ | $Div_k(\rightarrow)$ | $Div_g(\rightarrow)$ | MPJPE($\downarrow$) | MPJVE($\downarrow$) | BAS($\rightarrow$) | $FID_{cd}(\downarrow)$ | $Div_{cd}(\rightarrow)$ | BED($\rightarrow$) |
| Ground Truth | – | – | 10.86 | 7.82 | – | – | 0.1791 | – | 12.53 | 0.5308 |
| $S = [1.0, 1.0]$ | 8.09 | 10.37 | 10.74 | 7.69 | 155.30 | 17.59 | 0.2198 | 18.15 | 9.82 | 0.3470 |
| $S = [1.0, 1.2]$ | 6.17 | 8.10 | 10.79 | 7.74 | 138.68 | 16.29 | 0.2154 | 13.66 | 10.55 | 0.3709 |
| $S = [1.0, 1.5]$ | 5.77 | 7.70 | 10.80 | 7.75 | 134.78 | 15.90 | 0.2091 | **13.40** | **10.65** | 0.3835 |
| $S = [1.2, 1.2]$ | **5.57** | 7.63 | 10.82 | 7.76 | 132.99 | 15.68 | 0.2031 | 14.17 | 10.58 | 0.3863 |
| $S = [1.2, 1.5]$ | 5.62 | **7.29** | 10.83 | 7.79 | **132.20** | **15.43** | 0.1978 | 18.57 | 10.14 | 0.3901 |
| $S = [1.5, 1.5]$ | 6.07 | 7.62 | **10.85** | **7.81** | 135.27 | 15.62 | **0.1961** | 24.56 | 9.68 | **0.3925** |

and interaction quality. For instance, increasing the guidance weights (*e.g.*, $S = [1.2, 1.5]$) improves reconstruction metrics like MPJPE and MPJVE, but at the cost of harming the interaction quality ($FID_{cd}$ increases significantly to 18.57). Conversely, high guidance on both scales ($S = [1.5, 1.5]$) improves diversity ($Div_k$, $Div_g$) but leads to the worst interaction scores ($FID_{cd}$ of 24.56). Our chosen setting, $S = [1.2, 1.2]$ (highlighted), provides an optimal balance, achieving the best $FID_k$ and strong reconstruction quality while maintaining competitive interactive metrics. This demonstrates that LDCFG enables a nuanced tuning of realism against interaction coherence.

Furthermore, because our representation disentangles body components, we can apply LDCFG for targeted artistic control. Fig. 8 visualizes this capability, showcasing independent guidance applied to the upper body, lower body, and their combination to create distinct stylistic motion variations while maintaining plausible interactions.

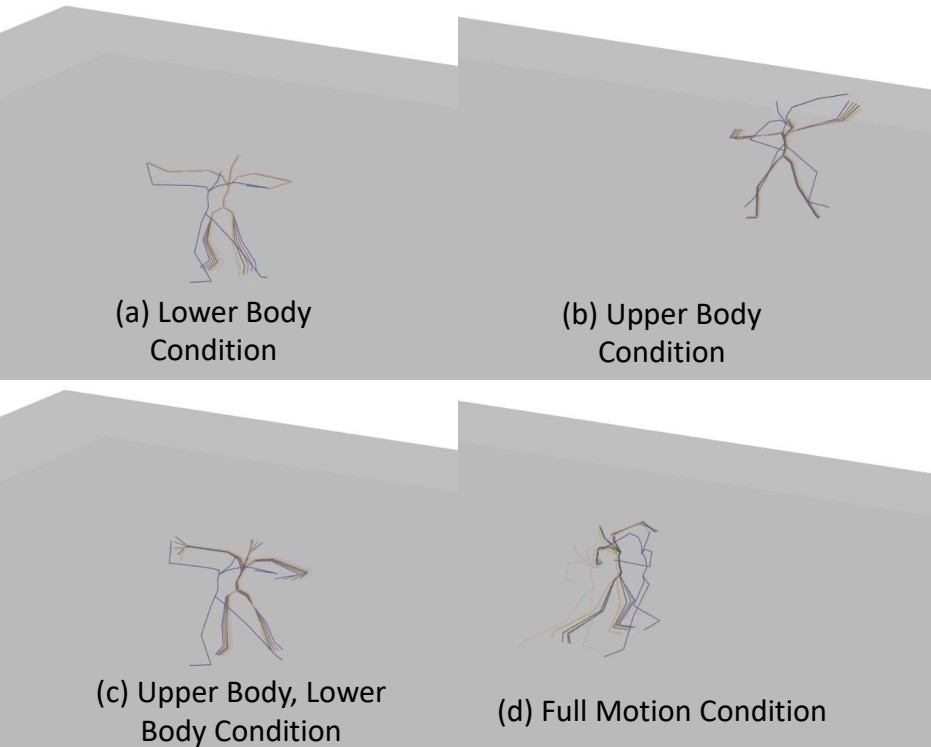

(a) Lower Body Condition

(b) Upper Body Condition

(c) Upper Body, Lower Body Condition

(d) Full Motion Condition

Figure 8: **Qualitative visualization of LDCFG's body-part control.** We apply decoupled guidance with varying strengths to different body regions. The blue robot is the leader. The other colored robots are generated by our model, each with a different LDCFG setting applied to the (a) lower body, (b) upper body, (c) their combination, or (d) full body global motion demonstrating fine-grained artistic control.

## A.5 USER STUDY RESULTS

Figure 9: **User study results.** ReactDance consistently outperforms all baselines across all criterions.

## A.6 MORE RESULTS ON RESIDUAL STAGES $R$ AND SCALABILITY ANALYSIS

In this section, we provide a detailed scalability analysis regarding the depth of our Hierarchical Finite Scalar Quantization (HFSQ), specifically the number of residual stages $R$. The choice of $R$ involves a critical trade-off between the representational capacity of the tokenizer and the modeling complexity for the subsequent diffusion generator.

We evaluate the performance across $R \in \{1, 2, 3, 4\}$ in Table 5. The results reveal three key observations:

- **Reconstruction Fidelity:** Increasing $R$ from 1 to 2 yields a substantial reduction in reconstruction error (MPJPE decreases from $35.53$ to $24.32$), indicating that the second stage is essential for capturing fine-grained articulation. However, further increasing $R$ to 3 or 4 provides diminishing returns (MPJPE only improves marginally to $22.97$) while linearly increasing the latent space size.

- **Generation Quality:** Surprisingly, higher stages ($R \geq 3$) lead to a degradation in generation quality ($FID_k$ increases from $5.57$ to $6.10$). We attribute this to the increased complexity of the latent distribution; as the sequence of residual codes becomes longer, the joint distribution becomes harder for the diffusion model to learn effectively given the same training budget.

- **Computational Cost:** The training time increases only marginally with $R$, rather than scaling proportionally. Specifically, $R = 2$ incurs a negligible overhead compared to $R = 1$ (1.00x vs. 0.94x) while offering significantly better motion quality.

Based on this analysis, $R = 2$ represents the optimal performance-efficiency trade-off, achieving the best balance between reconstruction fidelity, generative realism, and computational efficiency. Consequently, we adopt $R = 2$ as our default setting.

Table 5: **Scalability Analysis of Residual Stages** $R$. While higher $R$ marginally improves reconstruction, $R = 2$ achieves the best generation quality (FID) and the optimal trade-off between performance and cost.

| Setting | Reconstruction Metrics (HFSQ) | | | Generation Metrics (Diffusion) | | | Training Cost |
| --- | --- | --- | --- | --- | --- | --- | --- |
| | $FID_k\downarrow$ | MPJPE$\downarrow$ | $FID_{cd}\downarrow$ | $FID_k\downarrow$ | $FID_g\downarrow$ | $FID_{cd}\downarrow$ | Train Time |
| $R = 1$ | 2.69 | 35.53 | 13.67 | 13.10 | 10.15 | 24.50 | **0.94x** |
| $R = 2$ **(Ours)** | 0.40 | 24.32 | 5.20 | **5.57** | **7.63** | **14.17** | 1.00x |
| $R = 3$ | 0.38 | 24.08 | 2.35 | 5.92 | 7.85 | 14.50 | 1.07x |
| $R = 4$ | **0.30** | **22.97** | **2.10** | 6.10 | 8.02 | 14.85 | 1.12x |

## A.7 ADDITIONAL ANALYSIS ON DSW STRIDE SELECTION

In Section 4.3, we validated the impact of training stride $s$ on interaction coherence. Here, we extend this ablation to include **Motion Continuity** (measured by Boundary Jitter) and **Training Cost** (relative training time), providing a justification for our design choice.

We report the comparison across a wide range of strides in Table 6. We define **Jitter** as the mean L2 velocity change at block boundaries (half-radius of 30 frames) to quantify the smoothness of the stitched motions.

**Analysis of Motion Continuity.** As the stride $s$ decreases (density increases), the model receives more frequent supervision on boundary transitions. Consequently, the Jitter score consistently improves ($18.08 \to 15.19$), confirming that the Dense Sliding Window (DSW) mechanism effectively acts as a phase-agnostic kinematic smoother. Notably, sparse strides ($s \geq 64$) fail to capture these transitions, resulting in severe discontinuities ($> 17.9$) and collapsed interaction quality ($\text{FID}_{cd} > 39$).

**Efficiency vs. Quality Trade-off.** While the theoretical dense limit ($s = 1$) achieves the lowest Jitter (15.19) and best $\text{FID}_{cd}$ (14.02), it incurs a substantial computational cost (**4.00**× training time compared to $s = 4$). In contrast, our default setting ($s = 4$) achieves comparable performance (Jitter 15.66, $\text{FID}_{cd}$ 14.17) while maintaining high training efficiency. The marginal gains of $s = 1$ do not justify the quadrupled training overhead. Thus, $s = 4$ represents the optimal performance-efficiency trade-off for scalable reactive dance generation.

Table 6: Full Ablation of Dense Sliding Window Stride ($s$). **Jitter** measures boundary discontinuity.

| DSW Stride Setting | Motion Continuity Jitter ($\downarrow$) | Interaction Quality $\text{FID}_{cd}$ ($\downarrow$) | Training Cost Train Time ($\downarrow$) |
|---|---|---|---|
| $s = 360$ (Sparse) | 18.08 | 146.22 | 0.01× |
| $s = 240$ (No Overlap) | 17.18 | 121.65 | 0.02× |
| $s = 64$ | 17.99 | 39.50 | 0.06× |
| $s = 16$ | 16.27 | 22.69 | 0.25× |
| $s = 4$ (**Ours**) | 15.66 | 14.17 | 1.00× |
| $s = 1$ (Extreme Dense) | **15.19** | **14.02** | 4.00× |

## A.8 HAND MOTION MODELING ANALYSIS

The DD100 dataset contains inherent high-frequency noise and jitter in the ground truth hand motions (*Noisy DD100 GT_Hand Video*), and conventional models often overfit to this noise during training, resulting in unstable generation. While baseline methods struggle to generate coherent reactive dances even without hand motions (see *Duolando Failure Video*), the inclusion of fine-grained hand modeling further exacerbates their performance degradation.

In contrast, our investigation reveals that ReactDance acts as a robust *motion denoiser*. This capability stems from the quantization mechanism of our Hierarchical Finite Scalar Quantization (HFSQ). By projecting continuous motion into discrete scalar codebooks, HFSQ effectively filters out high-frequency artifacts while preserving meaningful articulation. As shown in Table 7, ReactDance achieves significantly lower Jitter (**25.32**) compared to the autoregressive baseline Duolando (**32.04**). Visual comparisons are provided in the *Hand Modeling Comparison Video*.

Table 7: **Hand Motion Analysis.** ReactDance effectively filters out high-frequency artifacts inherent in the dataset, significantly outperforming Duolando in motion stability (lower Jitter).

| Method | MPJPE $\downarrow$ | Jitter $\downarrow$ | $\text{FID}_g \downarrow$ |
|---|---|---|---|
| Duolando | 340.40 | 32.04 | 9.37 |
| **ReactDance (Ours)** | **244.23** | **25.32** | **8.85** |

A.9    ABLATION OF CONDITIONING SIGNALS

We investigate the distinct roles of the leader's motion and the musical accompaniment in guiding the generation process. We trained variants of our model by removing each condition individually. The results are summarized in Table 8.

- **Effect of Leader Motion:** Removing the leader's motion causes a collapse in interaction quality, as evidenced by the sharp degradation in $FID_{cd}$ ($14.2 \rightarrow 48.8$). This confirms that the leader's signal is critical for establishing the structural and spatial framework of the reaction.
- **Effect of Music:** Interestingly, removing the music condition leads to a slight improvement in MPJPE (reconstruction error) but a significant degradation in interaction realism ($FID_{cd}$ $14.2 \rightarrow 35.4$). This phenomenon occurs because, without music, the model becomes a "mechanical tracker", tending to generate safe poses to minimize geometric error. However, it loses the nuances required for high-fidelity dance generation, confirming that music is essential for fine-grained interactions.

Table 8: **Ablation of Conditioning Inputs.** Results indicate that the leader's motion is crucial for structural interaction, while music is essential for fine-grained interactive realism.

| Method | Solo Metrics | | Interactive Metrics | |
|---|---|---|---|---|
| | MPJPE ↓ | MPJVE ↓ | $FID_{cd}$ ↓ | BAS → |
| Ground Truth | - | - | - | 0.1791 |
| w/o. Leader Motion | 238.50 | 22.76 | 48.82 | 0.2074 |
| w/o. Music | **130.27** | **15.05** | 35.35 | 0.1947 |
| **ReactDance (Full)** | 132.99 | 15.68 | **14.17** | 0.2031 |

A.10    GENERALIZATION VIA ROLE SWAPPING

To verify that ReactDance learns generalized interaction rules rather than memorizing fixed leader-follower pairs, we conducted a **Role Swapping** experiment. Specifically, we inverted input conditions by feeding follower's motion as the leader signal and tasked the model to generate the leader's part.

As demonstrated in *Role Swapping Video*, the model successfully generates valid, semantically coherent reactions for the reversed pairs. This confirms that ReactDance captures the underlying physical and choreographic rules of interaction (*e.g.*, synchronization, relative spacing) independent of specific character roles.

A.11    ZERO-SHOT GENERALIZATION ON OUT-OF-DISTRIBUTION DATA

To assess the robustness of ReactDance beyond the specific choreography of the training set (DD100), we conducted a **Zero-Shot** evaluation using the **FineDance** dataset (Li et al., 2023). FineDance contains diverse solo dance genres that are disjoint from our training data, serving as rigorous Out-of-Distribution (OOD) input for the leader.

Despite the significant domain shift in dance styles and musical accompaniment, ReactDance synthesizes physically plausible and semantically coherent reactions. This suggests that the model has successfully learned generalized interaction dynamics—such as velocity matching, spatial awareness, and rhythmic synchronization—rather than merely memorizing dataset-specific motion patterns.

Qualitative demonstrations of this generalization capability are provided in following videos:

- **Spatial Precision:** The model maintains precise upper limbs positioning relative to the unseen leader (*Video: Upper Limbs Spatial Alignment*).
- **Dynamic Synchronization:** The reactor successfully anticipates and matches complex rotational movements (*Video: Synchronized Spinning*).
- **Musical Generalization:** The model generates appropriate rhythmic responses to unseen musical styles (*Video: Rhythmic Accompaniment*).
- **Kinematic Adaptability:** The model handles extreme kinematic variations, adapting to unseen bending leader poses (*Video: Large Pose Adaptation*).

