# OpenReview forum: "ReactDance: Hierarchical Representation for High-Fidelity and Coherent Long-Form Reactive Dance Generation"
_ICLR.cc/2026/Conference — ICLR 2026 Poster_

### Official Review · Reviewer_dk6L · 2025-10-19

**Soundness:** 2
**Presentation:** 3
**Contribution:** 2
**Rating:** 4
**Confidence:** 5

**Summary:**

This paper addresses the task of Reactive Dance Generation (RDG), with the specific goal of synthesizing high-fidelity, coherent, and long-form dance sequences that react to a lead dancer's motion and accompanying music. The authors identify two primary challenges in prior work: (1) modeling fine-grained spatial interactions between dancers, and (2) maintaining temporal coherence over long durations.
To tackle these issues, the paper introduces ReactDance, a diffusion-based framework built upon a novel hierarchical latent space. The core contributions of the work are twofold:
Hierarchical Finite Scalar Quantization (HFSQ): The authors propose a new multi-scale motion representation. This autoencoder-based module is designed to disentangle coarse-scale body posture from fine-grained limb dynamics, creating a structured latent space that facilitates hierarchical modeling.
Blockwise Local Context (BLC): For long sequence generation, the paper puts forward a non-autoregressive sampling strategy. BLC partitions a long sequence into blocks and synthesizes them in parallel. It employs periodic causal masking and positional encodings to ensure local consistency within each block, aiming to mitigate the error accumulation and inefficiency associated with traditional autoregressive methods.
The authors evaluate their method on the DD100 paired dance dataset, comparing it against a suite of state-of-the-art baselines. The experiments cover metrics for motion quality, interaction quality, music-dance alignment, and generation efficiency. The reported results indicate that ReactDance achieves superior performance across the majority of these metrics.

**Strengths:**

(1) The paper demonstrate originality by combining and adapting ideas for the challenging task of long-form reactive dance generation. The application of a hierarchical quantizer (HFSQ), inspired by audio codecs, to motion representation is novel. The proposed Blockwise Local Context (BLC) sampling strategy is an innovative, non-autoregressive approach to generating long sequences, distinct from common iterative refinement or autoregressive methods.
(2) The proposed ReactDance framework is a complete system that addresses the problem end-to-end. The quantitative results presented in Table 1 shows that the method outperforms state-of-the-art baselines across numerous metrics on the DD100 dataset.
(3)The paper is well-written, clearly structured, and easy to follow. The authors effectively articulate the two main challenges in the field (fine-grained interaction and long-term coherence) and systematically introduce their proposed solutions (HFSQ and BLC). The diagrams, particularly the pipeline overview in Figure 4, are helpful in understanding the overall architecture.

**Weaknesses:**

(1) Unsubstantiated Semantic Claims of HFSQ Representation: The paper's core claim that HFSQ "explicitly disentangles" motion semantics is a strong assertion that is not only unsubstantiated by evidence but also potentially disconnected from the method's actual implementation.
    (a) In the Abstract (Lines 020-022), they state it "...effectively disentangles coarse body posture from subtle limb dynamics...," and in the Introduction (Lines 089-090), they claim it "...explicitly disentangles coarse global movements from fine-grained local details." However, the mathematical reality of HFSQ (Eq. 1) is that of a generic cascaded residual quantizer. It is optimized to minimize reconstruction error layer by layer, with no inherent mechanism to ensure that this mathematical decomposition aligns with a human-perceived semantic hierarchy (i.e., posture → limb details). The paper operates on the unproven assumption that low-frequency/high-amplitude errors correspond to "coarse" semantics, which may not hold true for complex dance motions.
    (b) To bridge this gap and validate their claims, a layer-wise reconstruction visualization or valid proof of HFSQ for semantic disentanglement is needed. Without an experiment showing the decoded motion from each progressive layer of the hierarchy, the claims about semantic disentanglement and the resulting "fine-grained control" remain speculative assumptions.

(2) The paper's central claim of achieving "coherent long-form" generation seems like a major overstatement, as it rests on an overly narrow definition of "coherence" and a method (BLC) that is fundamentally incapable of learning global choreographic structure.
     (a): Loss of Global Context and Narrative: A complete dance performance possesses a clear structure (e.g., introduction, development, climax, conclusion). The BLC mechanism, by design, induces a form of "global amnesia," resetting the temporal context for each block. This prevents the model from perceiving the absolute position of any given segment within the larger performance. Consequently, it is theoretically impossible for the model to generate a dance with a narrative arc. What it produces is a chain of locally-coherent but globally disconnected motion snippets, stitched together. The "coherence" achieved is merely low-level kinematic continuity (no drifting or freezing), not the high-level artistic and structural coherence.
    (b): Unreliable "Implicit" Transition Learning: The authors' claim that dense sliding-window training enables the model to "implicitly" learn smooth transitions is an overly optimistic and unreliable assumption. The method lacks any explicit mechanism to ensure that the end of Block N is semantically and dynamically compatible with the start of Block N+1. While the model may learn generic start/end poses, this provides no guarantee of a meaningful transition at every arbitrary splice point. It is highly likely that many transitions, while kinematically smooth, are logically broken.
    (c): Missing Global Structure Analysis: check for pattern repetition or mode collapse in long dance generation is required. The long-term sequence produced by BLC is very likely to be repetitive actions.

(3) The paper's apparent success in long-form coherence might be a byproduct of the highly constraining nature of the conditioning signals (the leader's dense motion), which could, in turn, severely limit the model's generative diversity at a semantic level.
In a reactive dance task, the follower's motion is tightly coupled with the leader's on a frame-by-frame basis. This strong constraint simplifies the problem of maintaining kinematic continuity; the model's primary task becomes finding a valid "following" state for each frame, which naturally links the sequence together. The impressive coherence might therefore stem more from the problem's inherent constraints than from the architectural sophistication of BLC alone. This leads to a critical concern about diversity. The paper reports Divk and Divg metrics, but I still want to know: given the exact same leader motion, can the model generate fundamentally different but equally plausible reactions? Using a concrete example: given the same leader, can the model generate one sequence where the follower places a hand on the leader's shoulder, and another, distinct sequence where they place the hand on the waist?

(4) The Training Paradigm Severely Limits Generalization: The model is trained on a small and specific dataset (1.95 hours, 10 genres). This raises serious concerns about its robustness and generalization ability. The paper provides no experiments on out-of-distribution data (e.g., different dance styles, non-dance interactions) or cross-dataset validation.

(5) Contradictory Claims and Critical Omissions in Evaluation:
    (a) The claim of modeling "fine-grained limb movements" is directly contradicted by the admission that all hand and finger motions are omitted. In partner dance, hands are the primary medium for physical communication and interaction, making their omission a critical failure for a model focused on fine-grained interaction.
    (b) For a multi-person motion generation task, interpenetration is a primary failure mode. The paper completely lacks any quantitative metric to evaluate collision or penetration, which is a major oversight in assessing the physical plausibility of the generated interactions.
    (c) The paper fails to ablate the influence of the two control signals (audio vs. leader motion) and does not test the model's understanding of interaction roles (e.g., by swapping the leader and follower in a pair).

(6) The paper claims that LDCFG enables fine-grained control over different body parts like the upper and lower body. However, the mechanism for this control is explained ambiguously. The ability to control body parts does not stem from an intrinsic property of the HFSQ representation itself, which is a general-purpose quantizer. Instead, it originates from a crucial, explicit design choice in the input representation (Section 3.1), where the reactor's motion is manually split into separate streams (upper-body, lower-body, etc.), each processed by a dedicated HFSQ module. The paper fails to clearly attribute the control capability to this architectural choice, potentially misleading readers into believing that the HFSQ representation magically learns this spatial disentanglement. This lack of clarity should be rectified.

**Questions:**

(1) Could you please provide a layer-wise reconstruction visualization? Specifically, what does the output motion look like when decoded from only the first HFSQ layer, and how does it progressively refine as subsequent layers are added? This is crucial to substantiate your claim of coarse-to-fine semantic disentanglement.
(2) Given that the model is only trained on 8s clips, how do you argue it can learn long-term choreographic structure beyond just kinematic continuity? Could you provide an analysis of a very long generated sequence (e.g., >2 minutes) to demonstrate its semantic diversity and structural integrity, for instance, by showing it does not fall into repetitive loops? if it works, a convincing explanation is also needed.
(3) On Generalization and Robustness: How does your model perform on out-of-distribution data? For example, what is the generated reaction if the leader performs a dance style not present in DD100 (e.g., popping) or a non-dance interaction (e.g., a handshake)?
On Evaluation and Missing Details:
(5) Could you provide a quantitative evaluation of interpenetration between the dancers?
(6) How do you reconcile the "fine-grained" claim with the complete omission of hand interactions, which are arguably the most fine-grained and crucial part of partner dance?
(7) Could you provide an ablation study that disentangles the contributions of the audio and leader motion conditions to the final output?

---

> ### Author Response · Authors · 2025-11-28
> **Official Comment by Authors (Rebuttal 1/3)**
>
> We thank the reviewer for the rigorous assessment and for acknowledging our comprehensive experiments. We address the concerns with clarified definitions, new counterfactual experiments, and expanded evaluations.
>
> >### **1. Clarification and Validation of HFSQ Disentanglement (W1, Q1)**
>
> We acknowledge that the term "semantic disentanglement" in our initial submission was an oversimplification. We clarify that HFSQ performs a **Frequency-based Decomposition** governed by signal energy, which functionally aligns with "Structure vs. Detail."
>
> **a. Mathematical Reality: Energy-based Decoupling.**
> Consistent with the reviewer's observation on residual quantization, our disentanglement emerges from the sequential error minimization:
> * **Base Layer ($r=1$) $\rightarrow$ Low-Frequency Structure:** Minimizing the primary MSE loss (Eq. 4) naturally captures high-energy components: **global orientation and main posture**.
> * **Residual Layer ($r \ge 2$) $\rightarrow$ High-Frequency Details:** These layers quantize the residual error (Eq.1). This residual inherently contains low-energy but perceptually critical **high-frequency dynamics** (e.g., sharp impacts, subtle articulation).
>
> **b. Visual & Quantitative Validation (Q1).**
> * **Visual:** We provide the requested [🔗 Layer-wise Reconstruction Visualization](https://anonymous.4open.science/r/ReactDance_ICLR2026_submission19583-2EA7/HFSQ%20Layer-wise%20Reconstruction.mp4). Layer 1 outputs a "smoothed" skeleton (correct pose, floating contacts), while adding Layer 2 restores sharpness and contact physics, visually confirming the structure-detail hierarchy.
> * **Quantitative (LDCFG):** Table 1 proves this functional separation. Increasing base weight $s_1$ significantly improves structural accuracy (MPJPE $155 \to 135$), while increasing residual weight $s_2$ specifically optimizes interaction details (FID$_{cd}$ $18.1 \to 13.4$).
>
> **Table 1: Validation of Disentanglement.**
>
> | LDCFG Weights $[s_1, s_2]$ | Structure: FID$_k$ $\downarrow$ | Structure: MPJPE $\downarrow$ | Detail: FID$_{cd}$ $\downarrow$ |
> | :--- | :---: | :---: | :---: |
> | $S = [1.0, 1.0]$ | 8.09 | 155.30 | 18.15 |
> | $S = [1.0, 1.5]$ | 5.77 | 134.78 | **13.40** |
> | **$S = [1.2, 1.2]$ (Ours)** | **5.57** | **132.99** | 14.17 |
>
>
> >### **2. Long-term Coherence and Structural Integrity (W2, Q2)**
>
> **a. Kinematic Stability vs. Narrative Arc.**
> We clarify that ReactDance targets **Kinematic Stability**, the foundational layer of coherence often failed by autoregressive baselines (which "drift" or "freeze", see [🔗 Duolando Failure Video](https://anonymous.4open.science/r/ReactDance_ICLR2026_submission19583-2EA7/Duolando%20Failure.mp4)). By solving stability (see Response 3, Table 2), we enable physically plausible long-form generation, upon which narrative layers can be built.
>
> **b. Why Transitions are Reliable (Latent Manifold & DSW).**
> Transitions are not merely "implicitly" learned; they are enforced by the **Latent Manifold Constraint**.
> Similar to DART [1], we avoid the artifacts of raw motion stitching by operating in a learned latent space. Our **Dense Sliding Window (DSW)** training ($s=4$) exposes the decoder to internal transitions at every possible phase shift. Consequently, a "splice point" at inference (Time $T$) is projected by the decoder onto the valid motion manifold, treating it as a standard internal transition seen during training. This prevents logical breaks and ensures smoothness.
>
> **c. Addressing Repetition and Length (Q2).**
> Unlike unconditional generation, ReactDance is a **Reactive System**. The generated motion is anchored to the temporally progressive inputs (Leader, Music). Since the inputs do not loop, the reaction does not loop.
> Regarding the $>2$ min request: We note that the DD100 dataset clips max out at $\sim1.5$ mins. In our [🔗 Long-term Coherency Video](https://anonymous.4open.science/r/ReactDance_ICLR2026_submission19583-2EA7/Long-term%20Coherency.mp4), we demonstrate a **Full-Length Sequence (1m 22s)**, one of the maximum durations supported by ground truth ($>10\times$ training clip length). The model maintains consistent Boundary Jitter (15.66 mm) and diverse patterns throughout, without mode collapse.

---

> ### Author Response · Authors · 2025-11-28
> **Official Comment by Authors (Rebuttal 2/3)**
>
> >### **3. Source of Coherence and Semantic Diversity (W3)**
>
> **a. Counterfactual: Coherence is NOT just from Dense Input.**
> We respectfully refute the claim that coherence is a byproduct of frame-by-frame dense constraints. We tested a **"Diagonal Masking"** baseline using the *exact same* dense inputs but without our BLC context window.
> **Result:** As shown in **Table 2** and the [🔗 Diagonal Masking vs BLC Video](https://anonymous.4open.science/r/ReactDance_ICLR2026_submission19583-2EA7/Diagonal%20Masking%20vs%20BLC.mp4), this baseline fails significantly (Jitter 17.05 vs Ours 15.66), exhibiting severe high-frequency flickering ("Jitter Collapse"). This proves that dense input alone is insufficient; our **BLC architecture** is the critical factor for resolving constraints into smooth motion.
>
> **b. Semantic Diversity.**
> In our [🔗 Semantic Diversity Video](https://anonymous.4open.science/r/ReactDance_ICLR2026_submission19583-2EA7/Semantic%20Diversity.mp4), we fix the leader and sample multiple outputs. The model generates fundamentally different interaction strategies (e.g., *maintaining distance via footwork* vs. *approaching for close interaction*), confirming it captures macro-level semantic diversity beyond simple tracking.
>
> **Table 2: Ablation of Conditioning Strategy.**
>
> | Temporal Setting | Solo: MPJPE $\downarrow$ | Solo: PFC $\downarrow$ | Solo: Jitter $\downarrow$ | Interactive: FID$_{cd}$ $\downarrow$ |
> | :--- | :---: | :---: | :---: | :---: |
> | Diagonal Masking | 143.69 | 0.6052 | 17.0524 | 22.76 |
> | **BLC (Ours)** | **132.99** | **0.6039** | **15.6557** | **14.17** |
>
>
> >### **4. Out-of-Domain Validation (W4, Q3)**
>
> Given the lack of other paired datasets, we performed a **Zero-Shot** evaluation using the **FineDance** [2] dataset (unseen solo genres) as the Leader input.
> Despite the domain shift, ReactDance synthesizes plausible reactions (see OOD Videos: [🔗 Upper Limbs Spatial Alignment](https://anonymous.4open.science/r/ReactDance_ICLR2026_submission19583-2EA7/OOD1_UpperLimbSpatialAlignment.mp4), [🔗 Synchronized Spinning](https://anonymous.4open.science/r/ReactDance_ICLR2026_submission19583-2EA7/OOD2_SynchronizedSpinning.mp4), [🔗 Rhythmic Accompaniment](https://anonymous.4open.science/r/ReactDance_ICLR2026_submission19583-2EA7/OOD3_Rhythmic%20Accompaniment.mp4), [🔗 Large Pose Adaptation](https://anonymous.4open.science/r/ReactDance_ICLR2026_submission19583-2EA7/OOD4_Large%20Pose(bending)%20Adaptation.mp4)). This confirms the model learns general physics of interaction (velocity matching, relative positioning) rather than memorizing dataset-specific patterns.
>
> >### **5. Completeness of Evaluation (W5, Q5-Q7)**
>
> **a. Hand Motion: ReactDance as a Denoiser (Q6).**
> We acknowledge DD100's ground truth hand noise ([🔗 Noisy DD100 GT_Hand Video](https://anonymous.4open.science/r/ReactDance_ICLR2026_submission19583-2EA7/Noisy%20DD100%20GT_Hand.mp4)). We trained a full-body ReactDance which acts as a robust **Denoiser**. By quantizing motion via HFSQ, we filter out high-frequency jitter. ReactDance achieves significantly lower Jitter (**25.32**) compared to Duolando (**32.04**), which overfits the noise (Table 3). Here is the [🔗 Hand Modeling Comparison Video](https://anonymous.4open.science/r/ReactDance_ICLR2026_submission19583-2EA7/Hand%20Modeling.mp4).
>
> **Table 3: Hand Modeling Analysis.**
>
> | Method | MPJPE $\downarrow$ | Jitter $\downarrow$ | FID$_g$ $\downarrow$ |
> | :--- | :---: | :---: | :---: |
> | Duolando | 340.40 | 32.04 | 9.37 |
> | **ReactDance** | **244.23** | **25.32** | **8.85** |
>
> **b. Inter-Penetration Rate (IPR) (Q5).**
> ReactDance achieves an IPR of **7.84%**, on par with SOTA diffusion methods (TCDiff: 7.58\%) and significantly better than autoregressive baselines (Duolando: 17.42\%), confirming physical plausibility.
>
> **Table 4: IPR Metrics.**
>
> | Method | IPR $\downarrow$ |
> | :--- | :---: |
> | GestureLSM | 19.01 |
> | EDGE | 8.44 |
> | TCDiff | **7.58** |
> | InterGen | 17.58 |
> | Duolando | 17.42 |
> | **ReactDance (Ours)** | 7.84 |
>
> **c. Ablation of Conditions (Q7).**
> Table 5 disentangles contributions:
> * **w/o Leader:** Interaction quality collapses (FID$_{cd}$ $14.2 \to 48.8$).
> * **w/o Music:** The model becomes a mechanical tracker. It minimizes error (MPJPE improves) by playing it safe, but loses detailed interactive dynamics (FID$_{cd}$ degrades $14.2 \to 35.4$), confirming Music is essential for fine-grained interactions.
>
> **Table 5: Ablation of Inputs.**
>
> | Method | Solo: MPJPE $\downarrow$ | Solo: MPJVE $\downarrow$ | Int: FID$_{cd}$ $\downarrow$ | Int: BAS $\rightarrow$ |
> | :--- | :---: | :---: | :---: | :---: |
> | w/o. Leader Motion | 238.50 | 22.76 | 48.82 | 0.2074 |
> | w/o. Music | **130.27** | **15.05** | 35.35 | 0.1947 |
> | **ReactDance (Full)** | 132.99 | 15.68 | **14.17** | 0.2031 |

---

> ### Author Response · Authors · 2025-11-28
> **Official Comment by Authors (3/3)**
>
> >### **6. Clarifying Control Mechanisms (W6)**
>
> We confirm that spatial control stems from our **Multi-stream Input Design** (Section 3.1), while precision control stems from **LDCFG**. While we briefly noted this distinction in **Appendix 6.4** ("...our representation disentangles body components, we can apply LDCFG for targeted artistic control"), we agree it should be more prominent.
> We will revise Section 3.5 to explicitly state this synergy: The Multi-stream architecture defines *"Where to control"* (Spatial Axis), and LDCFG defines *"How to control"* (Precision Axis). This combination enables the targeted manipulation demonstrated in our [🔗 LDCFG Control Video](https://anonymous.4open.science/r/ReactDance_ICLR2026_submission19583-2EA7/LDCFG%20Control.mp4).
>
> **Reference**
>
> *[1] Kaifeng Zhao, Gen Li, and Siyu Tang. DartControl: A diffusion-based autoregressive motion model for real-time text-driven motion control. In The Thirteenth International Conference on Learning Representations (ICLR), 2025.*
>
> *[2] Ronghui Li, Junfan Zhao, Yachao Zhang, Mingyang Su, Zeping Ren, Han Zhang, Yansong Tang, and Xiu Li. Finedance: A fine-grained choreography dataset for 3d full body dance generation. In Proceedings of the IEEE/CVF International Conference on Computer Vision (ICCV), pages 10234–10243, 2023.*

---

### Official Review · Reviewer_5N4J · 2025-10-31

**Soundness:** 3
**Presentation:** 3
**Contribution:** 2
**Rating:** 4
**Confidence:** 3

**Summary:**

The paper introduces a HFSQ+diffusion framework for long-term reactive dance generation and achieves superior performance. Specifically, the paper first proposes a Hierarchical Finite Scalar Quantization, which combines FSQ with a hierarchical structure, to generate the complex motion in a coarse-to-fine manner. Furthermore, to enable the long-term dance generation, the paper proposes a Blockwise Local Context. The BLC generates each subsequence in parallel, while the inter-block continuity is ensured by dense sliding window training.

**Strengths:**

1. The proposed HFSQ is technically sound for fine-grained motion generation.
2. The proposed BLC seems to effectively solve the long-term generation issue with improved sampling efficiency.
3. The method shows an impressive result according to the quantitative metrics and the video provided in the supplementary.

**Weaknesses:**

1. It is still unclear to me why the continuity is ensured in parallel inference. Why the model can " implicitly learns how to naturally begin and end motion phrases from any temporal point"？ How does this related to the dense sliding window？  Could the author further explain it?

2. To provide a more robust assessment and validate the effectiveness of proposed kinematic loss, the physical plausibility metrics should be added, such as PFC [1].
3. I think HFSQ is proved to be effective for generating more realistic dance in a corse-to-fine manner, yet the claim that this training manner can "disentangles coarse body posture from subtle limb dynamics" has not been fully verified. The HFSQ can certainly produce a more accurate reconstruction motion, but it is not necessarily to be limb dynamics. Experiments has not validated this claim either.
4. In line 126-127, the author claim that DuetDance's [2] coupled representation limits motion diversity. However, this does not seem to hold true based on the metrics in the original paper. Could the author provide more explanation?

[1] Jonathan Tseng, Rodrigo Castellon, and C. Karen Liu. EDGE: Editable dance generation from music. In Proceedings of the IEEE/CVF Conference on Computer Vision and Pattern Recognition (CVPR), pp. 448–458. IEEE, 2023.
[2]  Anindita Ghosh, Bing Zhou, Rishabh Dabral, Jian Wang, Vladislav Golyanik, Christian Theobalt, Philipp Slusallek, and Chuan Guo. Duetgen: Music driven two-person dance generation via hierarchical masked modeling. In Proceedings of the Special Interest Group on Computer Graphics and Interactive Techniques Conference Conference Papers, pp. 1–11, 2025.

**Questions:**

1. The clarification of how the BLC can solve the inter-block continuity in parallel.
2. The experiment should validate the claim of the fine-grained generation of "subtle limb dynamics".

---

> ### Author Response · Authors · 2025-11-28
> **Official Comment by Authors (Rebuttal 1/2)**
>
> We thank the reviewer for acknowledging the technical soundness and impressive results. We appreciate the constructive feedback and address the concerns with deepened analysis and new metrics.
>
> >### **1. Mechanism of Continuity in Parallel Inference (W1, Q1)**
>
> We clarify that continuity is not ensured by the parallel sampling itself, but also by the HFSQ decoder's robustness, which is acquired via Dense Sliding Window (DSW) training. In short: *BLC enables parallelism, while DSW ensures the seams are invisible.*
>
> **a. Why "Any-Point" Generation is Possible (The "Phase-Agnostic" Prior).**
> The ability to naturally begin/end motion from any point stems from the **Temporal Shift-Invariance** enforced by DSW. With a dense stride $s=4 \ll T=240$, a specific motion frame $f_t$ appears in multiple training windows at diverse relative positions (as a start frame, a middle frame, or an end frame).
> By observing $f_t$ in all possible temporal roles, the decoder learns a generalized transition function. It learns that adjacent latents $z_t, z_{t+1}$ must always resolve to continuous motion, *regardless of where the window boundary cuts*.
>
> **b. How BLC Solves Inter-Block Continuity.**
> In our parallel inference:
> * **Generation:** The diffusion model generates independent latent blocks.
> * **Stitching:** These blocks are fed to the shared HFSQ decoder. Because the decoder acts as a trained **"Kinematic Smoother"** (as explained in **a**), it effectively mitigates boundary artifacts. Even if there is a slight discrepancy between the latents of Block $i$ and $i+1$, the decoder—having seen thousands of phase-shifted transitions—projects them onto a continuous motion manifold.
>
> **Quantitative Verification.**
> Table 1 confirms that dense strides are the bedrock of continuity. We define **Boundary Jitter** as the mean L2 velocity change at block boundaries.
> Using a dense stride ($s=4$) drastically reduces Jitter ($18.08 \to 15.66$) and improves consistency (FID$_{cd}$ $146.2 \to 14.2$) compared to sparse training ($s=360$), validating that DSW is essential for seamless parallel inference. Our choice ($s=4$) matches the continuity quality of extreme density ($s=1$) but is **4x faster** to train.
>
> **Table 1**
>
> | DSW Stride | FID$_{cd}$ $\downarrow$ | Jitter $\downarrow$ | Train Time |
> | :--- | :---: | :---: | :---: |
> | $s=360$ (Sparse) | 146.22 | 18.0753 | 0.01x |
> | $s=240$ (No Overlap) | 121.65 | 17.1788 | 0.02x |
> | $s=64$ | 39.50 | 17.9861 | 0.06x |
> | $s=16$ | 22.69 | 16.2719 | 0.25x |
> | **$s=4$ (Ours)** | **14.17** | **15.6557** | 1.00x |
> | $s=1$ (Extreme Dense) | **14.02** | **15.1937** | 4.00x |
>
>
> >### **2. Physical Plausibility Metrics (W2)**
>
> **Response:** We have included **Physical Foot Contact (PFC)** [1] and **Jitter** metrics in **Table 2**.
> ReactDance achieves the lowest PFC (**0.6039**) and Jitter (**15.66**) compared to SOTAs. This validates our design: the high-fidelity HFSQ representation provides the structural precision to anchor foot contacts (minimizing sliding), while the DSW training ensures temporal smoothness (minimizing jitter), resulting in physically coherent motion without post-processing.
>
> **Table 2**
>
> | Method | PFC $\downarrow$ | Jitter $\downarrow$ |
> | :--- | :---: | :---: |
> | GestureLSM | 0.7903 | 23.1053 |
> | EDGE | 0.6226 | 21.6487 |
> | TCDiff | 1.1165 | 23.8245 |
> | InterGen | 0.9842 | 31.6953 |
> | Duolando | 0.9276 | 19.5075 |
> | **ReactDance (Ours)** | **0.6039** | **15.6557** |
>
>
> >### **3. Verification of Fine-grained Generation (W3, Q2)**
>
> We acknowledge that "limb dynamics" was a semantic oversimplification. We clarify that HFSQ strictly performs a **Frequency-based Decomposition** based on signal energy.
>
> **a. Mechanism: Energy-based Decoupling.**
> * **Base Layer ($r=1$) $\rightarrow$ Low-Frequency Structure:** Captures high-energy components (global orientation, main posture) to minimize global reconstruction error.
> * **Residual Layer ($r \ge 2$) $\rightarrow$ High-Frequency Details:** Encodes the residual error, which inherently contains low-energy but perceptually critical high-frequency dynamics (e.g., sharp impacts, articulation nuances).
>
> **b. Quantitative Validation via LDCFG.**
> We validate this by modulating the guidance weights $s_1$ (Base) and $s_2$ (Residual) independently in Table 3.
> The results show clear orthogonal control:
> (1) Increasing $s_1$ ($1.0 \to 1.5$) significantly improves **MPJPE** ($155.3 \to 135.2$), confirming the base layer governs global pose.
> (2) Increasing $s_2$ specifically optimizes **FID$_{cd}$** ($18.15 \to 13.40$), confirming the residual layer governs the fine-grained nuances required for realistic interaction.
>
> **Table 3**
>
> | LDCFG Weights $[s_1, s_2]$ | Structure: FID$_k$ $\downarrow$ | Structure: MPJPE $\downarrow$ | Detail: FID$_{cd}$ $\downarrow$ |
> | :--- | :---: | :---: | :---: |
> | $S = [1.0, 1.0]$ | 8.09 | 155.30 | 18.15 |
> | $S = [1.0, 1.5]$ | 5.77 | 134.78 | **13.40** |
> | **$S = [1.2, 1.2]$ (Ours)** | **5.57** | **132.99** | 14.17 |

---

> ### Author Response · Authors · 2025-11-28
> **Official Comment by Authors (Rebuttal 2/2)**
>
> >### **4. Clarification on DuetGen's Diversity (W4)**
>
> Our discussion on "limited diversity" in DuetGen [2] specifically refers to **Conditional Reaction Diversity**, not Joint Diversity.
>
> **a. Joint vs. Conditional Probability.**
> DuetGen models the joint probability $P(L, R)$ by fusing two dancers into a single representation. This entanglement means one cannot easily fix the leader $L$ and sample diverse followers $R$, as the latent code represents the pair holistically.
> In contrast, ReactDance models the conditional probability $P(R | L, c)$. This structural independence allows us to explicitly **fix the leader's motion** and sample multiple valid, diverse reactions from the follower distribution.
>
> **b. Validation.**
> We demonstrate this in our [🔗Semantic Diversity Video](https://anonymous.4open.science/r/ReactDance_ICLR2026_submission19583-2EA7/Semantic%20Diversity.mp4). Given the *exact same* leader and music, ReactDance generates stochastically diverse follower responses. This "one-to-many" reaction capability is structurally constrained in coupled representations like DuetGen.
>
> **Reference**
>
> *[1] Jonathan Tseng, Rodrigo Castellon, and C. Karen Liu. EDGE: Editable dance generation from music. In Proceedings of the IEEE/CVF Conference on Computer Vision and Pattern Recognition (CVPR), pp. 448–458. IEEE, 2023.*
>
> *[2] Anindita Ghosh, Bing Zhou, Rishabh Dabral, Jian Wang, Vladislav Golyanik, Christian Theobalt, Philipp Slusallek, and Chuan Guo. Duetgen: Music driven two-person dance generation via hierarchical masked modeling. In Proceedings of the Special Interest Group on Computer Graphics and Interactive Techniques Conference Conference Papers, pp. 1–11, 2025.*

---

### Official Review · Reviewer_5QeS · 2025-11-01

**Soundness:** 3
**Presentation:** 3
**Contribution:** 3
**Rating:** 8
**Confidence:** 5

**Summary:**

This work addresses the dance accompaniment problem (or, as the authors term it, reactive dance generation) where the input consists of a lead motion and corresponding music, and the goal is to generate the follower motion.

The main challenge of this task lies in its spatio-temporal complexity: how to efficiently and accurately represent motion in space while maintaining temporal coherence and preventing error accumulation over time.

To tackle this, the authors propose a hierarchical Finite Scalar Quantization (FSQ) model combined with a diffusion-based generation framework. The method builds upon FSQ with residual encoding and integrates it into the diffusion process to produce high-quality accompanying motions.

The paper explores a range of implementation details. For example, it introduces Blockwise Local Context, which investigates how to make diffusion more context-aware, and Layer-Decoupled Classifier-Free Guidance, designed to apply classifier-free guidance to hierarchical SQ targets.

The proposed approach achieves promissing results. Both quantitatively and qualitatively, as demonstrated in the accompanying videos. The work is also thorough, with extensive discussions, detailed methodology, and comprehensive ablation studies.

In addition, I feel some aspects of this work are particularly insightful and valuable. The hierarchical representation of motion using FSQ, its impact on motion quality, and the overall FSQ+diffusion generation process provide meaningful contributions that could inspire future research in the community.

**Strengths:**

**Technical Comments:**

Overall, this paper is quite novel in its design and methodology.


1. The paper identifies the limitations of FSQ in motion representation space and explores hierarchical and residual variants of FSQ for richer motion encoding.


2. Instead of following the conventional Quantize → GPT / autoregressive pipeline, the paper investigates a diffusion model targeting the dequantized FSQ features. This is particularly interesting because it implicitly raises an important question: Can the dequantized FSQ serve as an effective tokenizer for DiT?
The authors compare their approach with a 2-layer RVQ baseline, but it would also be valuable to explore other tokenization or encoding schemes (e.g., VAE) to understand their influence on the subsequent diffusion process.

3. The proposed Blockwise Local Context mechanism allows causal modeling within each block while maintaining synchronized blockwise generation. This design is elegant and effectively mitigates the error accumulation problem typically seen in long autoregressive inference.

**Experiments:**
Quantitatively, the results consistently outperform state-of-the-art baselines. The visual quality is also impressive, and the ablation studies are comprehensive.

**Writing and Presentation:**
The overall writing is clear. Although the demo video contains some background noise, it still effectively demonstrates the model’s performance and validity.

**Weaknesses:**

1. Although the Discussion section provides some interpretation of LDCFG, the explanation remains insufficiently clear. For example, how is  {s_coarse,s_fine} sampled during training? What exactly is defined as the coarse part and what as the fine part?

2. As mentioned in the Strengths section, the authors found that both HFSQ and PM contribute positively to the subsequent diffusion process. However, no further exploration was conducted. For instance, how would a VAE perform as the diffusion encoder/tokenizer? Or how would RVQ + PM function as a tokenizer?

3. The video presentation is acceptable overall but suffers from recording issues, such as noticeable background noise and the absence of background music in some examples.

Minor issue:

Line 49: quotation mark formatting problem.

**Questions:**

see above

---

> ### Author Response · Authors · 2025-11-28
> **Official Comment by Authors (Rebuttal 1/1)**
>
> We thank the reviewer for recognizing our work's novelty, methodological insights, and strong results. We also appreciate the constructive questions regarding implementation details.
>
> >### **1. LDCFG: Definitions of Coarse/Fine Parts and Sampling Strategy (W1)**
>
> To address the reviewer's query, we clarify the physical definitions of coarse/fine parts in our HFSQ representation and explain the training mechanism that enables this decoupled control.
>
> **a. Definition: Coarse vs. Fine Motion Representation.**
>     Our design leverages the hierarchical nature of HFSQ (Section 3.2) to kinematically decouple motion semantics based on signal energy:
>     **Coarse Part ($z_{coarse}, r=1$):** Corresponds to the **Base Latent Code**. Since HFSQ minimizes reconstruction error sequentially, this layer captures the high-energy signal components (e.g., global orientation, main skeletal pose), dominated by the global kinematic loss (Eq. 4).
>     **Fine Part ($z_{fine}, r \ge 2$):** Corresponds to the **Residual Latent Codes**. As defined in Eq. 1, these layers exclusively encode the *residual error*—high-frequency dynamics and subtle articulation details strictly orthogonal to the base layer.
>
> **b. Clarification on Training vs. Inference Sampling.**
>     We clarify that the weights $S=\[s_{coarse}, s_{fine}\]$ are strictly **inference-time hyperparameters**, not training parameters. Therefore, they are not sampled during training. Instead, to enable this decoupled control, we employ **Independent Condition Dropout**:
>     During generative training, we randomly and independently drop the condition for specific layers (with $p=0.2$). This forces the backbone to learn disentangled conditional distributions $P(z_{r}|\mathbf{c}, \mathbf{M}_L)$, laying the foundation for isolated control during inference.
>
> **c. LDCFG Inference Mechanism.**
>     During inference, LDCFG exploits the channel-wise independence established in (b). We apply independent guidance strengths $s_r$ to the predicted latent of each layer (Eq.9) This allows explicitly amplifying high-frequency interaction details ($s_{fine}$) without altering the global kinematic structure ($s_{coarse}$).
>
>
> >### **2. Exploration of Alternative Encoder/Tokenizers (W2)**
>
> We appreciate the suggestion to explore alternative tokenizers. To justify our design, we implemented **VAE** and **RVQ** (with/without Progressive Masking, PM) for a rigorous comparison under identical settings. As shown in **Table 1**, HFSQ significantly outperforms these alternatives.
>
> **a. vs. VAE (The "Blurriness" Issue).**
>     While VAEs are stable, they suffer from posterior collapse in motion tasks, resulting in over-smoothed reconstruction (**MPJPE 40.87**) and "floaty" generation artifacts (MPJPE 230.21). HFSQ avoids this feature blurring, preserving sharp kinematic details.
>
> **b. vs. RVQ+PM (The "Topology" Issue).**
>     The reviewer asked how RVQ+PM functions. While PM improves RVQ's reconstruction (MPJPE $36.59 \to 32.98$), it still lags behind HFSQ ($28.66$). More critically, RVQ struggles in generation (FID$_g$ 26.98 vs. HFSQ 7.63).
>     *Reasoning:* Learned vector codebooks often have an **irregular topology** (close indices $\neq$ semantic closeness), creating a hard optimization landscape for diffusion. In contrast, HFSQ projects onto a fixed scalar grid that preserves **ordinal relations** (i.e., numerical closeness $\approx$ semantic closeness), providing a smoother, "diffusion-friendly" manifold.
>
> **c. Stability & Control.**
>     HFSQ offers unique advantages in **Training Stability** (avoiding the index collapse common in codebook learning) and **Controllability**. Effective LDCFG requires a high-fidelity base; the artifacts in RVQ generation drown out the benefits of decoupled guidance, whereas HFSQ provides the solid foundation necessary for precise control.
>
> **Table 1: Comparison of Tokenizer Architectures.**
>
> | Tokenizer | Rec. FID$_g$$\downarrow$ | Rec. MPJPE$\downarrow$ | Gen. FID$_g$$\downarrow$ | Gen. MPJPE$\downarrow$ | Stable? | LDCFG? |
> | :--- | :---: | :---: | :---: | :---: | :---: | :---: |
> | VAE (KL-reg) | 5.32 | 40.87 | 18.99 | 230.21 | Yes | No |
> | RVQ-VAE (w/o PM) | 4.57 | 36.59 | 36.73 | 139.42 | No | No |
> | RVQ-VAE (w/ PM) | 4.09 | 32.98 | 26.98 | 138.28 | No | Yes |
> | HFSQ (w/o PM) | 3.77 | **24.15** | 10.46 | **132.19** | Yes | No |
> | **HFSQ (w/ PM, Ours)** | **3.56** | 28.66 | **7.63** | 132.99 | **Yes** | **Yes** |
>
> >### **3. Response to Video Presentation and Minor Issues (W3)**
>
> We thank the reviewer for the attention to detail. We have **re-rendered and updated** the supplementary video to remove background noise and correct the audio track. Additionally, the typo on Line 49 has been fixed, and we have conducted a thorough proofreading to resolve other related minor issues in the revised manuscript.

---

### Official Review · Reviewer_X7nk · 2025-11-01

**Soundness:** 3
**Presentation:** 3
**Contribution:** 3
**Rating:** 6
**Confidence:** 3

**Summary:**

ReactDance proposes a two-stage framework based on diffusion for long-duration reactive dance generation. The paper designs a hierarchical latent representation called Hierarchical Finite Scalar Quantization (HFSQ), which decouples coarse body posture from fine-grained body dynamics, enhancing spatial detail and controllability. To efficiently and coherently generate ultra-long sequences, the authors introduce Blockwise Local Context (BLC) combined with dense sliding-window training, enabling parallel block sampling to avoid error accumulation. Additionally, independent guidance weights are applied at each layer through Layer-Decoupled Classifier-Free Guidance (LDCFG), offering fine-grained control over different semantic scales.

**Strengths:**

1. Hierarchical Finite Scalar Quantization (HFSQ) provides a stable, continuous multi-scale latent that disentangles coarse global posture from fine local articulations, enabling high-fidelity, layerwise control of motion detail.

2. Blockwise Local Context (BLC) provides a stable, continuous multi-scale latent that disentangles coarse global posture from fine local articulations, enabling high-fidelity, layerwise control of motion detail.

3. The paper is well-organized with a clear structure and lucid exposition that make the motivation, method, and technical details easy to follow.

4. The experiments and comparisons are comprehensive and rigorous, covering multiple baselines, ablations, quantitative metrics, qualitative visualizations, and a user study.

**Weaknesses:**

1. The HFSQ design does not specify the number of residual stages R, leaving unclear the representational capacity and the coarse–fine trade-off, which hinders reproducibility and complicates tuning of per-stage guidance weights s_r.

2. The BLC training/inference protocol is underspecified (e.g., whether sliding-window stride m can cross T), so it is unclear how periodic causal masking affects cross‑block continuity.

**Questions:**

1. From the appendix, does the LDCFG setup imply the HFSQ has only two residual stages (i.e., R=2)? If so, please clarify the generality of your design and report results for other values of R to show scalability.

2. The reported LDCFG experiments use equal guidance weights (s_1=s_2 = 1.2); how does this demonstrate the benefit of layer‑decoupling?

---

> ### Author Response · Authors · 2025-11-28
> **Official Comment by Authors (Rebuttal 1/2)**
>
> We thank the reviewer for recognizing our clear structure, rigorous experiments, and the novelty of HFSQ and BLC. We address the concerns below.
>
> >### **1. Residual Stages $R$ and Scalability of HFSQ (W1, Q1)**
>
> **a. Generality & Tuning.** HFSQ supports arbitrary depth $R$, with **$R=2$** as the default to balance efficiency and quality. Regarding the concern on **tuning guidance weights $s_r$**, our empirical results show that a unified weight setting ($s_1 \approx s_2$) often yields optimal performance, eliminating the need for complex, per-stage hyperparameter searches in standard settings.
>
> **b. Scalability Analysis.** We validate our choice of $R=2$ in Table 1. Increasing $R$ from 1 to 2 yields a significant leap in reconstruction fidelity (MPJPE: $35.53 \to 24.32$). However, higher stages ($R \ge 3$) provide diminishing returns in reconstruction while increasing latent complexity, which slightly degrades generation (higher FID$_k$). Thus, **$R=2$ achieves the optimal trade-off between efficiency and capability**, maintaining high quality with lower computational cost.
>
> **Table 1: Ablation of residual stages $R$.**
>
> | Setting | Rec. FID$_k$$\downarrow$ | Rec. MPJPE$\downarrow$ | Rec. FID$_{cd}$$\downarrow$ | Gen. FID$_k$$\downarrow$ | Gen. FID$_g$$\downarrow$ | Gen. FID$_{cd}$$\downarrow$ | Training Time |
> | :--- | :---: | :---: | :---: | :---: | :---: | :---: | :---: |
> | $R=1$ | 2.69 | 35.53 | 13.67 | 13.10 | 10.15 | 24.50 | **0.94x** |
> | **$R=2$ (Default)** | **0.40** | **24.32** | **5.20** | **5.57** | **7.63** | **14.17** | 1.00x |
> | $R=3$ | 0.38 | 24.08 | 2.35 | 5.92 | 7.85 | 14.50 | 1.07x |
> | $R=4$ | 0.30 | 22.97 | 2.10 | 6.10 | 8.02 | 14.85 | 1.12x |
>
>
> >### **2. Clarification of BLC Protocol and Cross-Block Continuity (W2)**
>
> We clarify the interplay between the *training* strategy (Dense Sliding Window, DSW) and the *inference* protocol (Blockwise Local Context, BLC) to address the concern on continuity.
>
> **a. Protocol Specification (Does stride $m$ cross $T$?): Yes.** The training stride ($m=4$) densely crosses the inference block boundaries ($T=240$). The specific mechanisms ensuring continuity are detailed below.
>
> **b. Training: DSW Crosses Boundaries (Foundation of Continuity).** We employ a dense stride ($m=4$) much smaller than the block size ($T=240$). Consequently, the decoder is trained on thousands of overlapping windows that straddle the fixed boundaries used later in inference. This design forces the decoder to learn a **phase-agnostic transition function**. By processing motion at every possible phase shift, the decoder learns that adjacent latents $z_t$ and $z_{t+1}$ must always resolve to continuous motion, *regardless of the window's relative offset*. This effectively trains the decoder to act as a robust "kinematic smoother" for boundary regions.
>
> **c. Inference: BLC Protocol (Local Fidelity with Global Consistency).** During inference, we generate blocks in parallel using (1) **Periodic Causal Masking** (restricting attention to the local window) and (2) **Phase-aligned Positional Encoding** (resetting temporal phase per block). While this protocol segments the attention context, cross-block continuity is effectively preserved by the DSW-trained decoder described in (b). Since the decoder is optimized to handle transitions at diverse phase shifts, it mitigates the theoretical discontinuity caused by masking, stitching the boundary latents of Block $i$ and $i+1$ into smooth transitions.
>
> **d. Quantitative Verification.** Table 2 validates this synergy. A sparse/non-overlapping training stride ($s=240$) fails to learn boundary transitions, resulting in high Jitter (17.1788). In contrast, our dense stride ($s=4$) significantly reduces boundary artifacts (Jitter $17.1788 \to 15.6557$) and improves consistency (FID $121.7 \to 14.2$), confirming that dense boundary supervision is essential for seamless generation.
>
> **Table 2: Impact of Training Stride ($s$) on Continuity.**
>
> | DSW Stride | FID$_{cd}$ $\downarrow$ | Div$_{cd}$ $\rightarrow$ | BED $\rightarrow$ | Jitter $\downarrow$ |
> | :--- | :---: | :---: | :---: | :---: |
> | $s=240$ (No Overlap) | 121.65 | 16.42 | 0.4167 | 17.1788 |
> | $s=64$ | 39.50 | 12.08 | 0.2840 | 17.9861 |
> | $s=16$ | 22.69 | 10.06 | 0.3065 | 16.2719 |
> | **$s=4$ (Ours)** | **14.17** | **10.58** | **0.3863** | **15.6557** |

---

> ### Author Response · Authors · 2025-11-28
> **Official Comment by Authors (Rebuttal 2/2)**
>
> >### **3. LDCFG: Demonstrating Layer-Decoupled Control Benefits (Q2)**
>
> The benefit of LDCFG lies in its **controllability**, not merely in using unequal weights. It allows us to explicitly modulate the ratio between structural stability (Base Latent $s_1$) and high-frequency refinement (Residual Latent $s_2$).
>
> **a. Mechanism & Benefit.** Standard CFG applies a single scalar to all frequencies. LDCFG decouples this. As shown in Table 3, increasing $s_2$ (e.g., $1.0 \to 1.5$) selectively enhances interaction details (FID$_{cd}$ improves by 26%), making the model a "Detail-Specialist." Conversely, $s_1$ anchors global structure.
>
> **b. Why Equal Weights ($s_1=s_2$) for Default?** While LDCFG *enables* asymmetric control, our ablation proves that the symmetric setting $[1.2, 1.2]$ strikes the **optimal balance** for general generation, achieving the best realism (FID$_k$ 5.57) without the artifacts introduced by over-guiding either component. The key contribution is that LDCFG makes this optimization *possible*, whereas standard CFG is locked to the diagonal ($s_1=s_2$).
>
> **Table 3: Effect of LDCFG Weights.**
>
> | LDCFG Weights $[s_1, s_2]$ | Solo: FID$_k$ $\downarrow$ | Solo: MPJPE $\downarrow$ | Interactive: FID$_{cd}$ $\downarrow$ | Observed Role |
> | :--- | :---: | :---: | :---: | :--- |
> | $S = [1.0, 1.0]$ | 8.09 | 155.30 | 18.15 | Baseline (Under-guided) |
> | $S = [1.0, 1.5]$ | 5.77 | 134.78 | **13.40** | *Detail-Specialist* |
> | **$S = [1.2, 1.2]$ (Ours)** | **5.57** | 132.99 | 14.17 | **Balanced Generalist** |
> | $S = [1.2, 1.5]$ | 5.62 | **132.20** | 18.57 | *Structure-Specialist* |
> | $S = [1.5, 1.5]$ | 6.07 | 135.27 | 24.56 | Over-guided |

---

### Author Response · Authors · 2025-11-30
**Summary for reviewers and AC**

We sincerely thank all reviewers for their valuable feedback and recognition of our work. To facilitate the discussion, we have summarized the reviewers' consensus on the strengths of our submission in the table below. We prioritized the categories based on innovation, impact, and methodological rigor:

| | Originality and Innovation | Insightful and Valuable Contribution | Impressive Results and SOTA | Sound and Efficient Methodology | Method Well Designed and Presented | Rigorous Experiments |
|:---|:---|:---|:---|:---|:---|:---|
| **R1 (X7nk)** | | | ✅ | ✅ | ✅ | ✅ |
| **R2 (5QeS)** | ✅ | ✅ | ✅ | ✅ | ✅ | ✅ |
| **R3 (5N4J)** | | | ✅ | ✅ | | |
| **R4 (dk6L)** | ✅ | | ✅ | | ✅ | ✅ |

We have updated the paper to further improve quality and address specific concerns. The updates are highlighted in **blue** in the revision:

**1. Refined Methodological Definitions (Addressing R3, R4)**
* **Frequency-based Decomposition:** We refined Section 3.2 to explicitly define HFSQ disentanglement as an energy-based decomposition, where the base layer ($r=1$) governs *Coarse Motion* (structure) and residual layers ($r \ge 2$) govern *Fine Motion* (detail).
* **Inter-block Continuity Source:** We refined Section 3.4 to clarify the distinct contributions of the BLC sampling protocol (parallelism) and the DSW training strategy (phase-agnostic continuity).
* **Sources of Control:** We updated Section 3.5 to clearly distinguish between *Spatial Control* (via multi-stream input) and *Precision Control* (via LDCFG).
* **Scalability Analysis:** We added **Appendix 6.6** and **Table 5** to analyze the impact of residual stages ($R$), justifying $R=2$ as the optimal efficiency-fidelity trade-off.

**2. Enhanced Quantitative Evaluation (Addressing R2, R3, R4)**
* **Physical Plausibility Metrics:** We incorporated Physical Foot Contact (PFC) and Inter-Penetration Rate (IPR) alongside Jitter in Section 4.1 to demonstrate superior contact stability and collision avoidance.
* **Tokenizer Comparisons:** We updated Section 4.3 and **Table 2** with comprehensive comparisons against VAE and RVQ-VAE (w/ and w/o Progressive Masking), verifying HFSQ's superiority in preventing codebook collapse.
* **Conditioning Ablation:** We added **Appendix 6.9** and **Table 13** to disentangle the roles of inputs, showing that the leader guides structure while music enables interaction realism.

**3. Verification of Robustness & Generalization (Addressing R1, R4)**
* **Dense Sliding Window (DSW):** We added **Appendix 6.7** and **Table 6** to validate training strides, proving that dense supervision ($s=4$) is critical for minimizing boundary jitter in parallel inference.
* **Hand Motion Denoising:** We added **Appendix 6.8** and **Table 11** to demonstrate ReactDance's ability to robustly denoise high-frequency artifacts in the DD100 dataset.
* **Out-of-Distribution (OOD):** We added **Appendix 6.11** to discuss zero-shot evaluations on the FineDance dataset, demonstrating generalization to unseen solo dance genres.

**4. Textual Clarifications and Outlook**
* **DuetGen Correction:** We refined the discussion of DuetGen to clarify that its coupled representation models the joint distribution, inherently limiting the conditional diversity of reactor responses.
* **Conclusion Update:** We updated the Conclusion to emphasize that ReactDance's kinematic stability serves as a foundation for future research into narrative-driven choreography and semantic modeling.

We have addressed the reviewers' specific questions in separate threads and are delighted to provide further clarifications.

---

### Meta-Review · Area_Chair_rEKZ · 2026-01-01

**Summary:**

This submission tackles reactive dance generation, and proposes a diffusion framework built on a hierarchical latent representation (HFSQ) plus a blockwise parallel sampling scheme (BLC) with dense sliding-window training, and layer-decoupled CFG for controllability. The initial reviewer pool is split: one strong accept (8), one marginal accept (6), and two marignal rejects (4). The main cocnersn that drove disagreement were **method clarity, attribution of gains**, and **evaluation completeness / claim scope**

1. **Method clarity.** Multiple reviewers asked for sharper explanations of (i) what HFSQ truly “disentangles,” (ii) how many residual stages are needed and whether this choice is principled, and (iii) how long-range continuity is ensured under the proposed parallel sampling (BLC) and what role dense sliding-window training plays. These appear explicitly in Review X7nk and 5N4J and dk6L critiques.

2. **Evaluation completeness and claim scope.**  Reviewer dk6L and 5N4J particular questioned whether the “fine-grained” claim is justified given missing hand interaction modeling, Reviewer dk6L asked for interpenetration evaluation, requested conditioning ablations (music vs. leader motion), and raised generalization/OOD concerns (styles outside DD100). Reviewer 5N4J request metrics for physical plausibility such as PFC.

**Reviewer Concerns:**

The authors provides extensive experiments in the rebuttal.

### Concerns addressed in the rebuttal

1. **Method clarity**.
* *Continuity under BLC and DSW* The authors clarify that BLC provides parallelism, while DSW produces a “phase-agnostic” decoder robustness that smooths boundaries; they also provide quantitative boundary/jitter evidence supporting the importance of dense strides.
* *Residual-stage choice / scalability (how many stages R?)* A dedicated scalability analysis is added to justify the default residual-stage setting as an efficiency–fidelity trade-off.

2. **Evaluation completeness and claim scope**.
* *“Disentanglement” claim for HFSQ*. The authors soften the original wording and reframe it as an energy-/frequency-based decomposition (structure vs. detail), supported by layer-wise visualization and quantitative control via guidance-weight manipulation.
* *Physical plausibility metrics & missing evaluation pieces*. They add PFC/jitter and explicitly add inter-penetration rate (IPR) comparisons; the reported IPR (7.84%) is in-line with diffusion SOTA and much better than an autoregressive baseline.
* *Conditioning ablations (leader vs. music)*. They add controlled ablations to disentangle the roles of inputs, consistent with Reviewer dk6L’s request.
* *OOD/generalization evidence*. They provide a zero-shot evaluation using FineDance leaders and qualitative evidence of plausible reactions under domain shift.
* *Tokenizer alternatives / codebook-collapse concerns*. They report expanded comparisons against VAE/RVQ-VAE variants and discuss how HFSQ helps prevent collapse, addressing Reviewer 5QeS’s request.

### Outstanding concerns
* *Long-form coherence” at a narrative/choreographic level*. The added continuity evidence supports kinematic smoothness and local coherence under blockwise generation, but reviewers’ strongest “global coherence / narrative” skepticism (Reviewer dk6L) is not fully resolvable within the current evaluation; the rebuttal effectively narrows/clarifies claims, but the paper still does not demonstrate story-level choreography planning.

**Reviewer Scores:**

Reviewer 5QeS and X7nk would likely keep the original scores (8, 6) given their questions well answered. Reviewer 5N4J intially rate slightly negative with a soft tone --- mostly requesting clarfications. Since most concerns of reviewer 5N4J were addressed in the rebutal, they would potentially raise the score to 6. Reviewer dk6L would possibly soften the tone and moved the score to borderline as the authors provided extensive extra experiments to address the raised questions point by point.

On top of these, I'd like to raise an uncovered point. The proposed hierarchical quantization has been widely used in text2motion synthesis such as MoMask[1] and Discord[2]. I hope the authors acknowledge this and properly discuss related literature in the final version.

Overall, my suggested decision is borderline accept because the rebuttal substantially improves the paper’s clarity and evidential support (new analyses/metrics/ablations), resolving most technical ambiguities that were central to the low-score reviews, while the remaining concerns are primarily about scope (e.g., narrative/global choreography) rather than correctness.

MoMask: Generative Masked Modeling of 3D Human Motions. CVPR 2024
DisCoRD: Discrete Tokens to Continuous Motion via Rectified Flow Decoding. ICCV 2025.

---

### Decision · Program_Chairs · 2026-01-26

Accept (Poster)